# FastTF: 4 Parameters are All You Need for Long-term Time Series Forecasting

## Abstract

Time series forecasting is essential across various sectors, including finance, transportation, and industry. In this paper, we propose FastTF, a powerful yet lightweight model in **T**ime-**F**requency domain for long-term time series forecasting. Our aim is to push the boundary of model lightweighting and facilitate the deployment of lightweight model on resource-constrained devices. Leveraging the global nature and information compressibility of the time series in frequency domain, we introduce patch-wise downsampling, Sparse Frequency Mixer (SFM), and patch predictor to capture the temporal variations of frequency components across different patches. Experimental results on five public datasets demonstrate that FastTF with very few parameters outperforms several state-of-the-art models and demonstrates a strong generalization capability. Notably, on the ETTh1 dataset, FastTF with only **4 parameters** achieves a performance that is close to the DLinear and FITS in the horizon-96 forecasting. Furthermore, we deployed our model on a FPGA development board (Zynq UltraScale+ RFSoC ZCU208 Evaluation Kit), where the corresponding resource usage statistics illustrate that our model has a very low computational overhead and latency, making it easily implemented on hardware devices.

## 1 Introduction

*"Less is more"* —— **Ludwig Mies van der Rohe**

Time series forecasting, a technique used to predict future data based on historical observations and has found extensive applications in various fields such as finance, transportation, energy, and meteorology. With the development of deep learning technologies, neural network-based models, including MLP (Zeng et al., 2023; Xu et al., 2023; Liu et al., 2024; Yi et al., 2024; Wang et al., 2024), CNN (Wu et al., 2022; Wang et al., 2023; Luo & Wang, 2024), and Transformer (Zhou et al., 2021; 2022; Nie et al., 2022), have gradually supplanted traditional models like ARIMA (Shumway et al., 2017) and have become mainstream. However, these models often face significant challenges: on one hand, their complexity makes them difficult to deploy on computationally constrained devices such as FPGAs or other embedded systems. On the other hand, as black-box models, they generally lack interpretability.

From the perspective of signal sequence characteristics, models only based on time-domain-based are effective at capturing trend information but fail to leverage the global nature and information compression capabilities of the frequency domain, leading to shortsightedness and high complexity. Conversely, models that only based on frequency domain are adept at capturing the inherent periodic characteristics within sequences but struggle to handle the sequential order of data, resulting in ambiguity in temporal information. To address these issues, we model the time series forecasting problem as one of intra-patch information fusion and inter-patch trend prediction. Specifically, the proposed FastTF model segments the time series into patches in the time domain and employs a downsampling strategy within patches, designed based on the Nyquist sampling theorem, to achieve efficient weight sharing across subsequences. Additionally, a Sparse Frequency Mixer (SFM) is introduced to address spectral leakage inherent in the discrete Fourier transform and to exploit the naturally sparse correlations between frequency points for intra-patch information fusion. The patch predictor is then used for inter-patch frequency trend prediction. This approach not only results in an exceptionally lightweight model that can be deployed on resource-constrained edge devices but

also provides a certain degree of interpretability. The contribution of this paper can be summarized as follows:

- We propose FastTF, a powerful yet extremely lightweight model for long-term time series forecasting.
- Based on the Nyquist sampling theorem, FastTF performs patch-wise downsampling and sub-sequence rFFT on time series data after dividing it into patches, maximizing weight sharing while minimizing information loss.
- Observing the sparse correlation between frequency points, we design SFM (Sparse Frequency Mixer), a linear mapping layer with a block diagonal weight matrix to facilitate information fusion between frequency points. The patch predictor then predicts the temporal variations of different frequency points and transform them back to the time domain.
- Experiments on five public datasets demonstrate that FastTF achieves superior performance compared to most mainstream models with remarkably low parameter counts and training overhead. Deployment results on an FPGA development board confirm that it can run with minimal resource and time overhead.

## 2 RELATED WORK

**Transformer Based Models** Transformers (Vaswani, 2017) stand out in time series forecasting due to their strong ability to capture long-range dependencies. For instance, Informer (Zhou et al., 2021) and Autoformer (Wu et al., 2021) capture the temporal dependencies of time series, while FEDformer (Zhou et al., 2022) models the frequency domain of time series. Recent study, like PatchTST (Nie et al., 2022), showed the effectiveness of patch-based processing in time series forecasting. However, these models are computationally expensive and suffer from a potential information loss due to the attention mechanism (Zeng et al., 2023).

**MLPs and CNNs** The latest research shows that simple linear models, like MLP and CNN, can achieve competitive performance in time series forecasting. For example, DLinear (Zeng et al., 2023), FITS (Xu et al., 2023) are two representative models that use single linear layers to capture the time and frequency characteristics of time series. ModernTCN (Luo & Wang, 2024), a CNN-based model that originated from modern convolution, also shows strong generalizability in time series forecasting.

## 3 PRELIMINARIES AND MOTIVATION

**DFT & FFT & rFFT** Given a discrete time sequence $x(n)$ with $n$ ranging from 0 to $N-1$, the Discrete Fourier Transform (DFT) converts this time-domain sequence into its frequency-domain representation. The DFT is defined as:

$$X(k) = \sum_{n=0}^{N-1} x(n)e^{-j2\pi kn/N} \tag{1}$$

where $X(k)$ is the value of the sequence at frequency index $k$, $N$ is the total number of samples in the sequence, and $e^{-j2\pi kn/N}$ is the complex exponential factor used to extract the frequency components of the sequence. FFT is a fast algorithm for computing the DFT which reduces the computational complexity from $O(N^2)$ to $O(N\log_2 N)$, making it feasible to compute the DFT for large sequences (Cooley & Tukey, 1965).

**Property 1.** *The DFT of a real-valued sequence $x(n)$ is Hermitian symmetric, i.e., $X(k) = X^*(N-k)$, where $X^*$ denotes the complex conjugate of $X$.*

Property 1 (with the proof given in the Appendix A.1) indicates that only around half of the frequency points are unique, and the other half can be obtained by taking the complex conjugate of the first half. Therefore, we use rFFT (Ziegler, 1972; Sorensen et al., 1987), which only computes around half of the frequency points to further reduce computational overhead. Frequency domain analysis is widely applied to time series and has the following characteristics:

- **Global Perspective:** As seen in equation 1, each frequency component in the DFT of a time series is related to all time indices, meaning each frequency component integrates information from the entire sequence.
- **Information Compression:** As illustrated in Figure 1, most time series in nature and daily life, after applying the DFT, exhibit characteristics of high and low-frequency components (shown in Figure 2). Most of the energy is concentrated in the low-frequency part, allowing us to filter out high-frequency noise and focus on the main components of the time series.

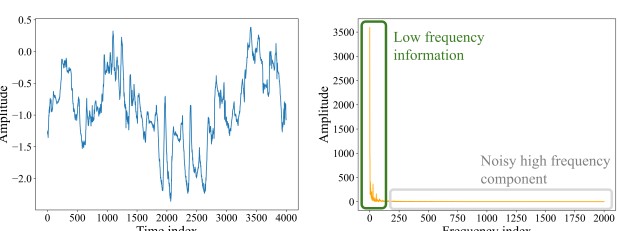 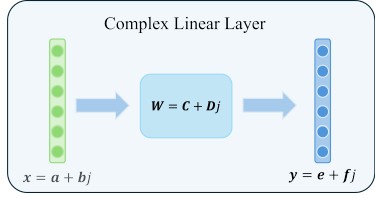

Figure 1: Original Time Series from *weather* dataset

Figure 2: The result of rFFT for time series in Figure 1

Figure 3: The structure of the *Complex Linear Layer*

On one hand, the **global perspective** of the frequency domain blurs time information, which drives us to seek a method that captures the variation of frequency over time. Specifically, this involves dividing the entire time series into several patches and predicting the changes in frequency points between different patches. On the other hand, the **compression of information** in the frequency domain inspires us to perform down-sampling and filtering within each patch, which will be detailed in Section 4.

**Complex Linear Layer** The Complex Linear Layer (Trabelsi et al., 2017) is a linear transformation layer that operates on complex numbers. Just as the linear layer in real-valued neural networks, given an $n$-dimensional input $\boldsymbol{x}$ and an $m$-dimensional output $\boldsymbol{y}$, the Complex Linear Layer is defined as $\boldsymbol{y} = \boldsymbol{W}\boldsymbol{x} + \boldsymbol{b}$, where $\boldsymbol{x} \in \mathbb{C}^n, \boldsymbol{y} \in \mathbb{C}^m, \boldsymbol{W} \in \mathbb{C}^{m \times n}, \boldsymbol{b} \in \mathbb{C}^m$ is the complex weight matrix, input, output, and bias, respectively (as shown in Figure 3). The complex linear layer has already been integrated into PyTorch implementations.

## 4 METHODOLOGY

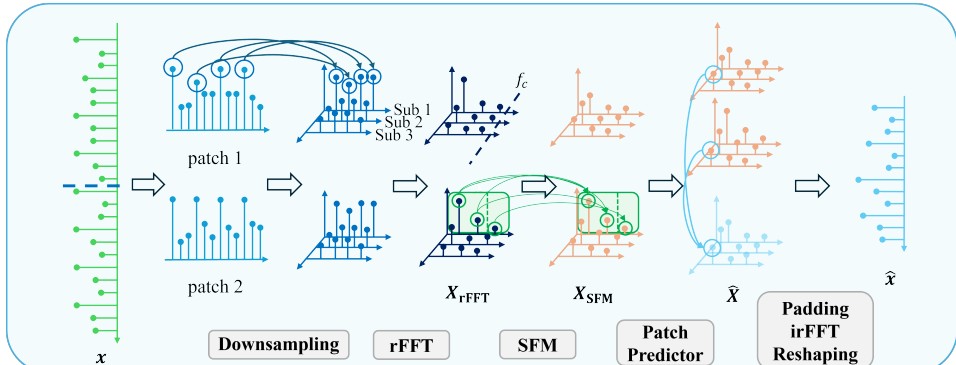

Figure 4: The architecture of FastTF

**Channel Independent Time Series Forecasting** Given a multivariate time series $\boldsymbol{X} = [\boldsymbol{x}^{(1)}, \boldsymbol{x}^{(2)}, \ldots, \boldsymbol{x}^{(N)}]$, where each $\boldsymbol{x}^{(i)}$ is a univariate series, the goal is to predict future values for each series independently. For each $\boldsymbol{x}^{(i)}$, given a historical window $\boldsymbol{x}^{(i)}_{t-L+1:t} \in \mathbb{R}^L$, we predict the future horizon $\hat{\boldsymbol{x}}^{(i)}_{t+1:t+H} \in \mathbb{R}^H$. For simplicity, we use $\boldsymbol{x}$ to represent $\boldsymbol{x}^{(i)}_{t-L+1:t}$ in the following sections.

The goal of this paper is to design a lightweight and efficient time series forecasting model. To achieve this, our approach is primarily based on three key ideas:

1. **Weight Sharing**: We promote weight sharing by splitting the complete sequence into multiple subsequences and applying the same operations to each subsequence or to the frequency points in parallel.

2. **Weight Sparsification**: The spectral leakage and the natural correlation (see Section 4.2) between frequency points in time series indicate sparse correlations between these frequency points, leading us to apply group sparsification to the weight matrix in the frequency domain.

3. **Patch-Scale Prediction**: To reduce the number of parameters, while at the same time capturing the temporal variations of frequency components, we predict the frequency points over an interval at the patch scale.

The architecture of our model, FastTF, is shown in Figure 4. FastTF first accepts a single-channel sequence $\boldsymbol{x} \in \mathbb{R}^L$ and divides it into $P$ patches, each of length $L/P$. Each patch is then down-sampled by a factor of $M$ and each subsequence is transformed into the frequency domain using rFFT, yielding $\boldsymbol{X}_{\text{rFFT}} \in \mathbb{C}^{P \times f_c \times M}$, where $f_c$ is the cut off frequency. The properly designed SFM (Sparse Frequency Mixer) is then applied to $\boldsymbol{X}_{\text{rFFT}}$ to capture the sparse correlations between frequency points in each subsequence, obtaining $\boldsymbol{X}_{\text{SFM}} \in \mathbb{C}^{P \times f_c \times M}$. The patch predictor gets $\boldsymbol{X}_{\text{SFM}}$ as input and predicts the future frequency components of each patch, yielding $\hat{\boldsymbol{X}} \in \mathbb{C}^{\frac{HP}{L} \times f_c \times M}$. Finally, the inverse rFFT (irFFT) and the reshape operation is applied to $\hat{\boldsymbol{X}}$ to obtain the predicted time series $\hat{\boldsymbol{x}} \in \mathbb{R}^H$. We now detail each component of FastTF in the following sections.

### 4.1 Patch-wise Downsampling and rFFT

**Theorem 1.** *Given a continuous-time signal $x(t)$ with a maximum frequency component $f_{max}$, sampling the sequence at a rate $f_s$ to get a descrete sequence $x(n)$ and downsampling it by a factor of $M$. To avoid spectral aliasing, the sampling frequency $f_s$ should satisfy the following condition:*

$$M \leq \frac{f_s}{2f_{max}},$$

*Proof.* See Appendix A.2. □

After patch division, a patch-wise downsampling strategy with a factor of $M$ is applied to each patch. Although Theorem 1 provide the necessary conditions to ensure that no information loss occurs during sampling, it must be noted that, in practice, signals (or sequences) that meet these criteria are virtually nonexistent. From a technical perspective, the sequence should first undergo low-pass filtering before downsampling to completely prevent spectral aliasing. However, considering that the high-frequency components of most sequences are negligible (as illustrated in Figure 2), and the additional computational cost associated with low-pass filtering (discussed in detail in Apppendix B), FastTF omits the filtering step and relies on empirical methods to select the value of $M$ in practice. Given downsampled sequences $\boldsymbol{X}_{\text{samp}} \in \mathbb{R}^{P \times \frac{L}{PM} \times M}$, we apply the rFFT to each patch and empirically select the cut-off frequency $f_c$ to obtain $\boldsymbol{X}_{\text{rFFT}} \in \mathbb{C}^{P \times f_c \times M}$. The overall process can be expressed as follows:

$$\boldsymbol{X}_{\text{rFFT}} = \text{Cut}\left(\text{rFFT}(\text{Downsample}(\boldsymbol{x}))\right) \tag{2}$$

where $\text{Cut}(\cdot)$ is a function that selects the first $f_c$ frequency points of the rFFT result.

### 4.2 Sparse Frequency Mixer (SFM)

The SFM (Sparse Frequency Mixer) is designed to integrate information between frequency points in the frequency domain. Specifically, the SFM is motivated by two key observations:

- **Spectral Leakage:** Proposition 1 (with the proof given in the Appendix A.3) and Figure 5a illustrate that for a single frequency signal, the sampled time sequence's energy can spread to surrounding frequency points due to the **mismatch of FFT frequency points.**

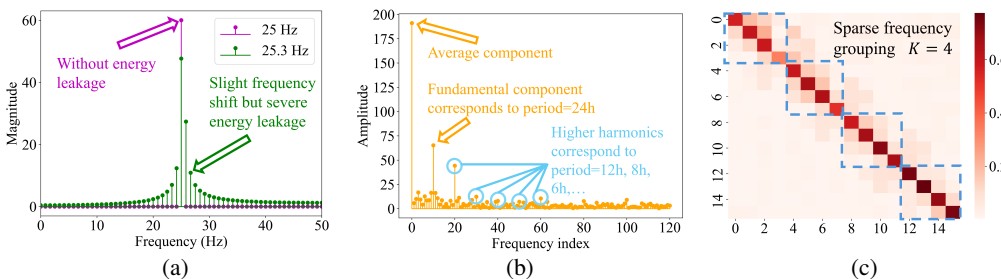

Figure 5: **Sparse correlation between frequency points:** (a) Spectral leakage phenomenon due to misalignment with DFT bins, the number of sampling points is 120 and the sampling rate is set to 100; (b) Higher harmonics generated by a base frequency; (c) The pattern of the weight matrix learned in the *Sparse Frequency Mixer* without sparsification.

- **Natual Correlation:** The frequency components of a time sequence can be naturally correlated, for example, the harmonics generated by a base frequency, as introduced in Proposition 2 (with the detailed explanation given in the Appendix A.4) and illustrated in Figure 5b.

**Proposition 1** (Spectral Leakage due to Misalignment with DFT Bins). *Let $x[n]$ be a discrete-time signal of length $N$, sampled at a rate $f_s$. The Discrete Fourier Transform (DFT) computes frequency components at specific frequency bins given by:*

$$f_k = \frac{k f_s}{N}, \quad k = 0, 1, \ldots, N - 1.$$

*If the signal $x[n]$ contains a sinusoidal component of frequency $f_0$, such that $f_0$ does not exactly match any of the DFT frequency bins $f_k$, i.e.,*

$$f_0 \neq f_k \quad \text{for any } k,$$

*then the energy of the sinusoidal component at $f_0$ will leak into adjacent frequency bins. This phenomenon is known as **spectral leakage**.*

**Proposition 2** (Harmonics). *Let $x(n)$ be a time series that is a periodic or quasi-periodic signal, possibly containing non-linearities or sharp transitions (e.g., discontinuities or sudden changes). The presence of such non-linearities or sharp transitions in $x(n)$ leads to the generation of higher harmonics in its frequency spectrum.*

These observations suggest that the frequency points in a time series are **correlated**, but the **correlation is sparse**, meaning that most frequency points only interact with a few nearby ones. This is further confirmed by a learned weight matrix of the frequency domain linear layer, which shows a sparse pattern. Specifically, weights connecting distant frequency points are nearly zero, as illustrated in Figure 5c. Based on these facts, we design the SFM to capture the sparse correlations between frequency points. For simplicity, we assume that the correlation between frequency points is block-diagonal, so that the weight matrix of the SFM can be also represented as a block-diagonal matrix, which is equivalent to a grouped linear layer. Specifically, given the input $\boldsymbol{X}_{\text{rFFT}}$, the SFM first reshapes the $f_c$ frequency points into $K$ groups, each containing $f_c/K$ frequency points, and then applies a linear transformation to each group for $PM$ rFFT sequences in parallel. Given SFM : $\mathbb{C}^{P \times f_c \times M} \to \mathbb{C}^{P \times f_c \times M}$, the operation of SFM can be formulated as:

$$\boldsymbol{X}_{\text{SFM}} = \text{SFM}(\boldsymbol{X}_{\text{rFFT}}) \tag{3}$$

$$= \text{Concat}(\mathcal{L}_1(\boldsymbol{X}_{\text{rFFT}}^{(1)}), \ldots, \mathcal{L}_K(\boldsymbol{X}_{\text{rFFT}}^{(K)})), \quad k = 1, 2, \ldots, K \tag{4}$$

where $\mathcal{L}_k(\cdot)$ is the linear transformation applied to the $k$-th group of frequency points, i.e., $\mathcal{L}_k : \mathbb{C}^{P \times \frac{f_c}{K} \times M} \to \mathbb{C}^{P \times \frac{f_c}{K} \times M}$. Concat$(\cdot)$ is a function that concatenates the results of the linear transformations, and $\boldsymbol{X}_{\text{rFFT}}^{(k)} \in \mathbb{C}^{P \times \frac{f_c}{K} \times M}$ denotes the $k$-th group of frequency points in $\boldsymbol{X}_{\text{rFFT}}$.

### 4.3 PATCH PREDICTOR AND POST-PROCESSING

The patch predictor is a linear layer that maps the mixed frequency components $\boldsymbol{X}_{\text{SFM}}$ to the future frequency components of each patch $\hat{\boldsymbol{X}}$ ($\mathbb{C}^{P \times f_c \times M} \to \mathbb{C}^{\frac{HP}{L} \times f_c \times M}$), and the weight is shared

across different frequency components. The predicted frequency components are then zero-padded to their original length $\left(\left\lfloor \frac{L}{2PM} \right\rfloor + 1\right)$ and transformed back to the time domain using irFFT followed by a reshape operation to obtain the final prediction $\hat{\boldsymbol{x}} \in \mathbb{R}^H$:

$$\hat{\boldsymbol{X}} = \text{PatchPredictor}(\boldsymbol{X}_{\text{SFM}}) \tag{5}$$

$$\hat{\boldsymbol{X}}_{\text{irFFT}} = \text{irFFT}\left(\text{ZeroPad}(\hat{\boldsymbol{X}})\right) \tag{6}$$

$$\hat{\boldsymbol{x}} = \text{Reshape}(\hat{\boldsymbol{X}}_{\text{irFFT}}) \tag{7}$$

where $\text{ZeroPad}(\cdot)$ is a function that zero-pads the frequency dimensioin, and $\text{Reshape}(\cdot)$ reshapes the irFFT result to obtain the final prediction ($\mathbb{C}^{\frac{HP}{L} \times \frac{L}{PM} \times M} \to \mathbb{R}^H$).

## 5 EXPERIMENTS

In this section, we first validate the performance of FastTF on five commonly used public datasets, followed by a detailed complexity analysis of FastTF, where we present a configuration framework with only 4 parameters. Next, we conduct a series of hyperparameter searches to analyze the performance under different hyperparameter settings. Moreover, experiments

Table 1: The statistics of the used forecasting datasets.

| Dataset | Traffic | Electricity | Weather | ETTh1 & ETTh2 |
|---|---|---|---|---|
| Channels | 862 | 321 | 21 | 7 |
| Sampling Rate | 1 hour | 1 hour | 10 min | 1 hour |
| Total Timesteps | 17,544 | 26,304 | 52,696 | 17,420 |

were conducted to verify the generalizability of FastTF on different datasets and horizons. Finally, we deployed FastTF on an FPGA development board and provided corresponding resource usage statistics to demonstrate its feasibility for deployment on hardware devices.

Table 2: **Comparison of different methods across various datasets and horizons.** The best 3 results are highlighted **red**, blue, *green*, respectively. Please note that all the results are reported after fixing the code bug (See Appendix D).

| Dataset | ETTh1 | | | | ETTh2 | | | | Electricity | | | | Traffic | | | |
|---|---|---|---|---|---|---|---|---|---|---|---|---|---|---|---|---|
| Horizon | 96 | 192 | 336 | 720 | 96 | 192 | 336 | 720 | 96 | 192 | 336 | 720 | 96 | 192 | 336 | 720 |
| FEDformer (2022b) | *0.375* | 0.427 | 0.459 | 0.484 | 0.340 | 0.433 | 0.508 | 0.480 | 0.188 | 0.197 | 0.212 | 0.244 | 0.573 | 0.611 | 0.621 | 0.630 |
| TimesNet (2023) | 0.384 | 0.436 | 0.491 | 0.521 | 0.340 | 0.402 | 0.452 | 0.462 | 0.168 | 0.184 | 0.198 | 0.220 | 0.593 | 0.617 | 0.629 | 0.640 |
| PatchTST (2023) | 0.385 | 0.413 | *0.440* | 0.456 | *0.274* | *0.338* | *0.367* | 0.391 | **0.129** | **0.149** | 0.166 | 0.210 | 0.366 | 0.388 | 0.398 | 0.457 |
| DLinear (2023) | 0.384 | 0.443 | 0.446 | 0.504 | 0.282 | 0.350 | 0.414 | 0.588 | *0.140* | 0.153 | *0.169* | 0.204 | 0.413 | 0.423 | 0.437 | 0.466 |
| FITS (2024) | 0.382 | *0.417* | 0.436 | 0.433 | 0.272 | 0.333 | 0.355 | 0.378 | 0.145 | *0.159* | 0.175 | *0.212* | *0.398* | *0.409* | *0.421* | 0.457 |
| U-Mixer(2024) | 0.370 | 0.423 | 0.470 | 0.500 | 0.290 | 0.366 | 0.423 | 0.446 | 0.151 | 0.163 | 0.179 | 0.210 | 0.451 | 0.458 | 0.477 | 0.520 |
| Koopa (2024) | *0.375* | 0.421 | 0.451 | *0.445* | 0.298 | 0.356 | 0.370 | 0.392 | 0.154 | 0.193 | 0.199 | 0.215 | 0.401 | 0.403 | 0.432 | *0.464* |
| MICN (2023) | 0.398 | 0.430 | *0.440* | 0.491 | 0.299 | 0.422 | 0.447 | 0.442 | 0.164 | 0.177 | 0.193 | *0.212* | 0.519 | 0.537 | 0.534 | 0.577 |
| **FastTF (ours)** | **0.350** | **0.388** | **0.419** | **0.416** | **0.268** | **0.330** | **0.351** | **0.376** | 0.132 | **0.149** | **0.165** | **0.200** | 0.387 | 0.400 | 0.410 | **0.446** |

### 5.1 EXPERIMENTAL SETUP

**Datasets** The five datasets used in our experiments follows Zhou et al. (2021). The statistics of the datasets are summarized in Table 1 and the detailed information are shown in Appendix E.

**Baseline** We compare FastTF with transformer-based models include FEDformer, and PatchTST, CNN-based models include TimesNet, MICN and ModernTCN, MLP-based models include DLinear, FITS, Koopa, U-Mixer (Ma et al., 2024) and TimeMixer. The baseline models are set to its best performance on each dataset. Please note that the look-back window for FastTF is set to 720. We use MSE (Mean Squared Error) as the evaluation metric and each model was evaluated on four different forecasting horizons: 96, 192, 336, and 720.

**Environment** We implement FastTF using PyTorch and train the model on a single NVIDIA RTX 4090 GPU.

Table 3: *Weather* dataset results.

| Dataset | Weather | | | |
|---|---|---|---|---|
| Horizon | 96 | 192 | 336 | 720 |
| FEDformer (2022b) | 0.217 | 0.276 | 0.339 | 0.403 |
| TimesNet (2023) | 0.172 | 0.219 | 0.280 | 0.365 |
| PatchTST (2023) | *0.149* | 0.194 | 0.245 | 0.314 |
| DLinear (2023) | 0.176 | 0.218 | 0.262 | 0.323 |
| FITS (2024) | 0.145 | 0.188 | 0.236 | 0.308 |
| TimeMixer (2024) | 0.147 | *0.189* | *0.241* | *0.310* |
| ModernTCN (2024) | 0.149 | 0.196 | 0.238 | 0.314 |
| Koopa (2024) | 0.154 | 0.193 | 0.245 | 0.321 |
| MICN (2023) | 0.161 | 0.220 | 0.278 | 0.311 |
| **FastTF (ours)** | **0.140** | **0.180** | **0.232** | **0.301** |

## 5.2 Main Results

Table 2 and Table 3 present the experimental results of FastTF and the baseline models on the five datasets. FastTF achieves competitive performance compared to the baseline models, with a significantly lower number of parameters. Specifically, FastTF achieved a nealy 10% of performance improvement compared to the MLP-based models like DLinear and FITS on horizon-96 forecasting on the *ETTh1* dataset. This is due to the fact that *ETTh1* dataset exhibits stronger seasonality and periodicity, which can be better captured by the Time-Frequency domain analysis of FastTF. Moreover, compared to complex models like FEDformer and TimesNet, FastTF achieves better performance, because the model better analized simple frequency shift and variation in the time series. Notably, FastTF performed well on both mid-term (like horizon-96 and horizon-192) and long-term forecasting tasks, which can be attributed to its patch prediction strategy. The hyperparameter settings and the additional results are provided in Appendix C.

## 5.3 Complexity Analisys

**Theorem 2** (parameter count). *Given the look-back window length $L$, the number of patches $P$, the output horizon $H$, and the cut-off frequency $f_c$, the number of frequency group $K$ in SFM, the total number of parameters in FastTF can be calculated as:*

$$Parameters = \frac{f_c{}^2}{K} + \frac{HP^2}{L} \tag{8}$$

*Specifically, without cutting off the frequency points, given the the down sampling factor $M$, the number of parameters in FastTF can be calculated as:*

$$Parameters = \left(\lfloor \frac{L}{2PM} \rfloor + 1\right)^2 / K + \frac{HP^2}{L} \tag{9}$$

*Proof.* The proof is provided in Appendix A.5 and the detailed parameter table is shown in Appendix G. The detailed complexity analysis can be found in Appendix H. ∎

Table 4: The parameter count of DLinear (2023), FITS (2024), and FastTF (ours) under different Horizon and Look-back settings on *ETTh1* dataset, where the patch size $L/P$ is set to 48 and the downsampling factor is set to 24.

| model | DLinear (2023) | | | | FITS (2024) | | | | **FastTF (ours)** | | | |
|---|---|---|---|---|---|---|---|---|---|---|---|---|
| Horizon \ Look-back | 96 | 192 | 336 | 720 | 96 | 192 | 336 | 720 | 96 | 192 | 336 | 720 |
| 96 | 18624 | 37056 | 64704 | 138K | 840 | 1,218 | 2,091 | 5,913 | **4** | 9 | 15 | 31 |
| 192 | 37248 | 74112 | 129K | 277K | 1,260 | 1,624 | 2,542 | 6,643 | 9 | 17 | 29 | 61 |
| 336 | 65184 | 130K | 226K | 484K | 1,890 | 2,233 | 3,280 | 7,665 | 15 | 29 | 50 | 106 |
| 720 | 140K | 278K | 485K | 1.04M | 3,570 | 3,857 | 5,125 | 10,512 | 31 | 61 | 106 | 226 |

Table 5: Comparison of various models in terms of parameters, maximum GPU memory usage, and inference batch time (with batch size 256 on cpu) on the *weather* dataset. For fair comparison, we set the look-back window to 720 and the predict horizon to 720.

| Model | Parameters | Max GPU Mem | Batch Time (cpu) |
|---|---|---|---|
| FEDformer (2022b) | 16.80 M | 6.05 G | 18.55s |
| MICN (2023) | 73.45 M | 4.97 G | 7401.7ms |
| ModernTCN (2024) | 176.18 M | out of memory | – |
| PatchTST (2023) | 8.7 M | 11.21 G | 2234.3ms |
| Koopa (2024) | 2.14 M | 12.614 G | 60.81ms |
| DLinear (2023) | 21.80 M | 660.80 M | 84.48ms |
| FITS (2024) | 951 K | 401.89 M | 68.16ms |
| **FastTF** (Ours) | **3.8 K** | **223.17 M** | **46.16ms** |

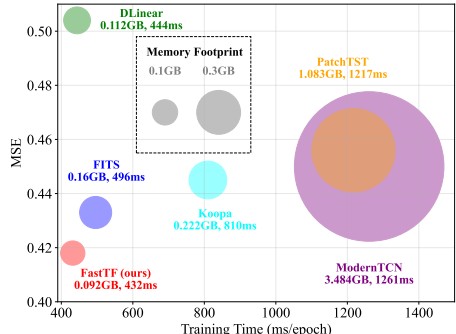

Figure 6: The memory footprint and training speed of different models on the *ETTh1* dataset. The look-back window and the predict horizon are both set to 720.

Theorem 2 provides the calculation of the number of parameters in FastTF. We then explore the limit of the number of parameters in FastTF by comparing the performance (MSE loss) with DLinear and

FITS on the *ETTh1* dataset, a small dataset with 7 similarly behaved channels. The result shown in Table 4 indicates that FastTF is able to conduct 96-96 forecasting with as few as **4** parameters, which is **210 times fewer than DLinear** and **4000 times fewer than that of FITS**. When it comes to 720-720 forecasting, FastTF achieves a comparable performance with only 226 parameters, which is nearly **5000 times fewer than DLinear**. As shown in memory footprint Figure 6, FastTF achieves the best performance while significantly reducing the memory usage and training time compared to other baselines. Specifically, FastTF is **38 times more memory efficient** and **3 times faster** than ModernTCN on the *ETTh1* dataset.

Furthermore, Table 5 shows the parameter count, GPU memory usage, and cpu inference time of FastTF and other models on the *weather* dataset, a dataset with 21 different behaved channels. Please note that a *individual* configuration is used for each model (See Appendix F for detailed information). Astonishingly, FastTF achieves the best performance with only **3.8K** parameters, which is up to **4400 times smaller** than the transformer-based models, and **46,400 times smaller** than the recent CNN-based models, ModernTCN. In short, FastTF is **lighter, faster, and better**.

### 5.4  4 Parameters are all you need−a case study on the ETTh1 dataset

**Proposition 3** (The ability to capture statistical property drift). *The rFFT operation can natually capture drift of statistical property in the time series, i.e., the variance of the mean value of the time series between different patches, bacause recall Equation 1 we have:*

$$X(0) = \sum_{n=0}^{N-1} x(n) = N \cdot mean(x), \qquad (10)$$

*which is also called the DC component of a spectrum.*

We now give a configuration framework on the *ETTh1* dataset for FastTF with only 4 parameters. Given the look-back window length $L = 96$, the number of patches $P = 2$, the cut-off frequency $f_c = 1$ (so that the 1 parameter in SFM can be omitted), the down sampling factor $M = 24$, and the number of frequency group $K = 1$, the performance of FastTF on the *ETTh1* dataset is shown in Table 6. This framework achieves a competitive perfor-

Table 6: The performance comparison of FastTF with only 4 and 8 parameters on the *ETTh1* dataset.

| Model (look-back length) | Param. count | MSE |
|---|---|---|
| FastTF (96) | **4** | 0.383 |
| FastTF(192) | 8 | **0.371** |
| FITS (720) | 5.9K | 0.382 |
| DLinear (720) | 138K | 0.384 |
| FEDformer (96) | 16.30 M | 0.380 |

mance compared to the baseline models, and reduced the number of parameters by **4,000,000 times** compared to FEDformer. Beneath this result lies an interesting coincidence: The points obtained by downsampling within a patch happen to be at the same position across two cycles (with the major cycle of *ETTh1* being 24). Proposition 3 indicates that when $f_c = 1$, the model captures the shift of mean value across patches by averaging each subsequence, essentially capturing the trend information on the scale of the entire cycle. The prediction results is shown in Figure 7.

### 5.5  Detailed Discussion

**The effect of downsampling** The downsampling within patches facilitates weight sharing, as subsequent SFM operates independently on each subsequence. However, the downsampling process lowers the maximum frequency that the rFFT can represent. Consequently, the high-frequency components of the original sequence's spectrum are aliased back into the low-frequency range, which results in irreversible information loss within individual

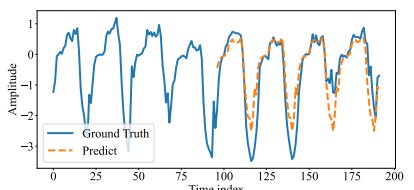

Figure 7: The prediction results of FastTF with only 4 parameters on the *ETTh1* dataset.

sequences. Fortunately, since the set of all subsequences essentially retains all the information from the original sequence, this information loss is partially compensated. We explore the extent to which this operation impacts the final model performance by varying the downsampling factor $M$, with the results presented in Table 7. The results indicates that although the performance usually decreases as the downsampling factor increases, the degradation is relatively marginal, which further demonstrates the robustness of FastTF. Interestingly, on the *weahter* dataset, the performance even improves when the downsampling factor is set to 24, which is consistent with the fact that the *weather* dataset has a relatively low frequency and is less sensitive to the downsampling operation.

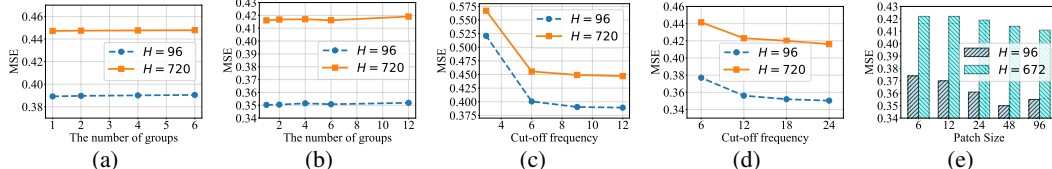

Figure 8: **Detailed study on hyperparameters.** (a) The effect of the number of frequency groups $K$ in SFM on the *traffic* dataset. (b) The effect of the number of frequency groups $K$ in SFM on the *ETTh1* dataset. (c) The effect of the cut-off frequency $f_c$ on the *traffic* dataset. (d) The effect of the cut-off frequency $f_c$ on the *ETTh1* dataset. (e) The effect of the patch size on the *ETTh1* dataset.

**The effect of SFM** The sparse grouping of frequency points in SFM reduce the number of parameters of the weight matrix linearly with the number of frequency groups $K$, but it also omits the potential long-range connections between frequency points. Therefore, we futher explore the effect of the number of frequency groups $K$ on the model performance on *ETTh1* and *traffic*, and the results are shown in Figure 8a and Figure 8b. The results imply that the increase in $K$ does not necessarily harm the model performance. In fact, the sparse grouping strategy can be seen as a form of regularization, which can prevent overfitting and improve the model's generalization ability. Moreover, the selection of cut-off frequency $f_c$ is also crucial. The results in Figure 8c and Figure 8d show that the model performance is relatively stable when $f_c$ is closed to $\frac{L}{2PM}$ and start to deteriorate rapidly when $f_c$ becomes too small. This is because most of the information in the time series is concentrated in the low-frequency range, and the information loss become significant when the cut-off frequency gets close to the main frequency of the time series. The result of the ablation study about the SFM is given in Appendix C.2.

**The effect of patch size** We explore the effect of the patch size on the model performance on the *ETTh1* dataset, and the results are shown in Figure 8e. The results suggest that a larger patch size is more advantageous for ultra-long-term forecasting, while a moderate patch size is beneficial for relatively short-term predictions. On the other hand, setting patch size to an integer multiple of the data's inherent periodicity also facilitates cross-period feature learning, thus improving the model's performance.

Table 7: Results obtained by varying the downsampling factor $M$ on three datasets, with the look-back window set to 672, and the patch size set to 48.

| Dataset | ETTh1 | | Weather | | Electricity | |
|---|---|---|---|---|---|---|
| Horizon | 96 | 720 | 96 | 720 | 96 | 720 |
| $M = 24$ | +0.012 | +0.007 | +0.009 | +0.004 | **-0.004** | +0.002 |
| $M = 8$ | +0.009 | +0.007 | +0.005 | +0.001 | +0.001 | +0.003 |
| $M = 4$ | +0.003 | +0.002 | +0.003 | +0.002 | +0.002 | +0.002 |
| $M = 2$ | +0.001 | **-0.002** | +0.002 | +0.000 | +0.001 | +0.003 |
| $M = 1$ | **0.350** | 0.418 | **0.144** | **0.306** | 0.145 | **0.204** |

## 5.6 GENERALIZABILITY

The generalization capability of a model, i.e., its ability to perform inference on other similar datasets after being trained on a specific dataset, is a crucial metric for evaluating the effectiveness of a method. Therefore, we evaluated FastTF on two transfer paths, *ETTh2* → *ETTh1* and *Electricity* → *ETTh1*, and presented the results in Table 8. The results demonstrate that FastTF significantly outperforms baseline models in both horizon of 96 and 720. Specifically, in 96 step, the *Electricity* dataset proves more suitable for transfer learning compared to *ETTh2*, while the opposite holds true for 720 step forecasts. This is because, in terms of long-term trends, *ETTh2* shares greater similarity with *ETTh1* (Lin et al., 2024). These findings indicate that FastTF can be effectively transferred to other datasets with minimal performance degradation, particularly for long-term predictions on *ETTh2*.

## 5.7 TRAINING DETAIL

In the final part of our experimental analysis, we focus on the training details. Figure 9 presents various training specifics of FastTF on the *ETTh1* dataset.

Figures 9a and 9b illustrate the training loss and validation loss curves of FastTF, along with a comparison to DLinear and FITS. The results indicate that FastTF converges more rapidly and with greater sta-

Table 8: The generalizability of FastTF on the ETTh2 → ETTh1 and Electricity → ETTh1 transfer paths.

| Dataset | ETTh2 → ETTh1 | | Electricity → ETTh1 | |
|---|---|---|---|---|
| Horizon | 96 | 720 | 96 | 720 |
| PatchTST | 0.452 | 0.478 | 0.405 | 0.473 |
| Koopa | 0.411 | 0.449 | 0.401 | 0.477 |
| DLinear | 0.422 | 0.526 | 0.390 | 0.469 |
| FITS | 0.414 | 0.446 | 0.388 | 0.452 |
| FastTF | **0.385** | **0.423** | **0.380** | **0.444** |

bility, with a significantly lower convergence value on the validation set compared to DLinear and FITS. In contrast, FITS and DLinear both exhibit unstable fluctuations during the early stages of training.

Figure 9c shows the predictive performance of FastTF. FastTF is able to better capture the peaks, troughs, and inherent periodicity within the sequence, demonstrating a significantly superior performance compared to DLinear and FITS.

Figure 9d visualizes the SFM weight matrix under the condition of $K = 3$. The three weight matrices are concatenated in a block-diagonal form, with the remaining parts displayed as zeros. Comparing this figure with Figure 5c, it can be observed that the primary portions of the weight matrix are preserved, while the less important weights at the edges are ignored. This linear reduction in the size of the weight matrix effectively retains the most critical information, consistent with the discussion in Section 5.5.

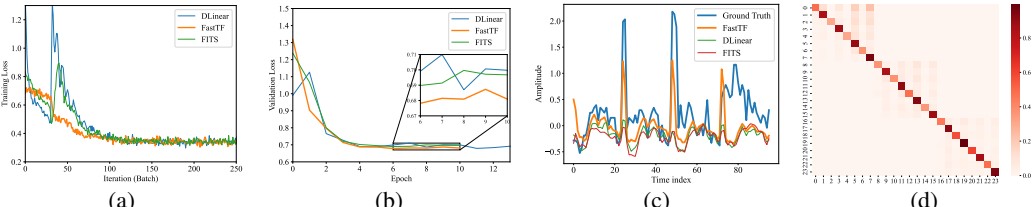

|   (a)   |   (b)   |   (c)   |   (d)   |

Figure 9: Training detail and visualization of FastTF on the *ETTh1* dataset. (a) Training loss curve, where learning rate is set to 0.008. (b) Validation loss curve. (c) Prediction results. (d) Visualization of the sparse weight matrix in SFM.

## 6 DEPLOYMENT ON FPGA DEVELOPMENT BOARD

FPGA (Field-Programmable Gate Array) is a versatile integrated circuit that enables users to program hardware according to specific requirements, facilitating the implementation of targeted functionalities. Owing to its advantages of low power consumption and low latency, FPGA is extensively utilized in digital signal processing applications. However, its limited storage and computational resources present challenges for the independent deployment of large neural network models. This paper focuses on the *Zynq UltraScale+ RFSoC ZCU208 Evaluation Kit* as a representative example to demonstrate the exceptional hardware deployability of FastTF. We conduct on-chip deployments of three algorithms—FastTF, DLinear, and FITS—comparing their resource consumption and latency in a 720-720 inference task. For FastTF, we set parameters at $M = 3$, $K = 2$, $f_c = 8$, and a patch size of 48, while the truncation frequency for FITS is set to 200. The results, summarized in Table 9, reveal that FastTF significantly conserves storage and computational resources while delivering faster predictions and lower operational power. Notably, both DLinear and FITS utilize more BRAM blocks and DSPs, which may be impractical for lower-configured FPGAs. Moreover, the inference time of FITS seems to be unreasonably high, which is due to its adoption of a serial computation scheme during the FFT stage, as the resource consumption of parallel algorithms far exceeds the limits of the development board. Please note that the results may vary depending on the implementation details. Additional details can be found in Appendix J.

Table 9: Resource Usage and Inference Time on FPGA. *Mul* denotes the number of real number multiplications (without considering rFFT).

## 7 CONCLUSION

In this paper, we proposed FastTF, a time series forecasting model that operates in the Time-Frequency domain. The model employs downsampling and inter-patch prediction to enable weight sharing, while frequency-domain weight sparsification further minimizes both parameter count and computational overhead. Through explanations and experimental results, we demonstrate that FastTF of-

| Metric | FastTF | DLinear | FITS |
|---|---|---|---|
| BRAMs | **5.5** | 722.5 | 407 |
| DSP Blocks | **810** | 2893 | 2410 |
| Power | **4.9W** | 5.1W | 5.9W |
| Cycles | **353** | 750 | 21851 |
| Mul | **20.5K** | 1054.8K | 240.0K |

fers seven key advantages: **simple, lightweight, fast, effective, robust, generalizable, and deployable**. These strengths strongly position FastTF as a highly competitive model for time series forecasting tasks, particularly on resource-constrained devices.

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

## A  PROOF

### A.1  PROOF OF PROPERTY 1

*Proof.* We want to prove that the Discrete Fourier Transform (DFT) of a real-valued sequence $x(n)$ is Hermitian symmetric, i.e., $X(k) = X^*(N - k)$.

The DFT of the sequence $x(n)$ is defined as:

$$X(k) = \sum_{n=0}^{N-1} x(n)e^{-j\frac{2\pi}{N}kn}$$

where $k = 0, 1, 2, \ldots, N - 1$.

The complex conjugate of $X(k)$, denoted as $X^*(k)$, is:

$$X^*(k) = \left( \sum_{n=0}^{N-1} x(n)e^{-j\frac{2\pi}{N}kn} \right)^*$$

Since the sum of the conjugates is equal to the conjugate of the sum, and $e^{-j\theta}$ is conjugated to $e^{j\theta}$, we have:

$$X^*(k) = \sum_{n=0}^{N-1} x^*(n)e^{j\frac{2\pi}{N}kn}$$

Given that $x(n)$ is a real-valued sequence, $x^*(n) = x(n)$. Thus:

$$X^*(k) = \sum_{n=0}^{N-1} x(n)e^{j\frac{2\pi}{N}kn}$$

Now, let's express $X(N - k)$ using the definition of the DFT:

$$X(N - k) = \sum_{n=0}^{N-1} x(n)e^{-j\frac{2\pi}{N}(N-k)n}$$

Simplifying the exponent:

$$X(N - k) = \sum_{n=0}^{N-1} x(n)e^{-j\frac{2\pi}{N}Nn}e^{j\frac{2\pi}{N}kn}$$

Since $e^{-j\frac{2\pi}{N}Nn} = e^{-j2\pi n}$ and $e^{-j2\pi n} = 1$ for any integer $n$, we have:

$$X(N - k) = \sum_{n=0}^{N-1} x(n)e^{j\frac{2\pi}{N}kn}$$

Notice that:

$$X^*(k) = \sum_{n=0}^{N-1} x(n)e^{j\frac{2\pi}{N}kn}$$

and:

$$X(N - k) = \sum_{n=0}^{N-1} x(n)e^{j\frac{2\pi}{N}kn}$$

Thus, we conclude:

$$X(N - k) = X^*(k)$$

We have shown that:

$$X(k) = X^*(N - k)$$

This result confirms that the DFT of a real-valued sequence $x(n)$ is Hermitian symmetric, completing the proof.

$\square$

### A.2   PROOF OF THEOREM 1

*Proof.* We begin by considering the original discrete-time sequence $x(n)$ which is sampled at a frequency $f_s$. The sequence $x(n)$ has a maximum frequency component $f_{\max}$.

The Nyquist sampling theorem states that to avoid aliasing, the sampling frequency $f_s$ must satisfy:

$$f_s \geq 2f_{\max}.$$

This ensures that the highest frequency component in the signal is adequately captured by the sampling process.

Now, consider downsampling the sequence $x(n)$ by a factor of $M$. Downsampling is the process of reducing the sampling rate by keeping every $M$-th sample and discarding the others. The new sequence after downsampling is denoted by $x_d(n) = x(Mn)$.

When we downsample by a factor of $M$, the effective sampling frequency after downsampling becomes:

$$f'_s = \frac{f_s}{M}.$$

To avoid aliasing in the downsampled sequence, the new sampling frequency $f'_s$ must satisfy the Nyquist condition:

$$f'_s \geq 2f_{\max}.$$

Substituting $f'_s = \frac{f_s}{M}$ into the inequality, we get:

$$\frac{f_s}{M} \geq 2f_{\max}.$$

Multiplying both sides by $M$, we obtain:

$$f_s \geq 2M \times f_{\max}.$$

This inequality indicates that for the downsampling process to avoid aliasing, the original sampling frequency $f_s$ must be at least $2M$ times the maximum frequency component $f_{\max}$ of the sequence $x(n)$.

Hence, the theorem is proved. $\square$

### A.3   PROOF OF PROPOSITION 1

*Proof.* The phenomenon of spectral leakage occurs when a sinusoidal component within a signal does not align perfectly with the frequency bins of the Discrete Fourier Transform (DFT). The DFT calculates the frequency content of a discrete-time signal $x[n]$ over a finite interval, producing frequency components at specific frequencies $f_k = \frac{kf_s}{N}$, where $k$ ranges from 0 to $N - 1$ and $f_s$ is the sampling frequency.

When the frequency of a sinusoidal component $f_0$ in the signal does not coincide with any of these discrete DFT frequency bins $f_k$, the DFT cannot represent $f_0$ as a single frequency component.

Instead, the energy that should ideally be concentrated at $f_0$ is distributed among several neighboring frequency bins, a phenomenon known as spectral leakage.

To understand why this occurs, consider the DFT of a sinusoidal signal $x[n] = A\cos(2\pi f_0 n + \phi)$. If $f_0$ aligns perfectly with one of the DFT bins $f_k$, the DFT will represent this component as a peak at $f_k$, with no energy in the other bins. However, if $f_0$ does not align with any $f_k$, we can express $f_0$ as $f_0 = f_k + \Delta f$, where $\Delta f$ is the mismatch between $f_0$ and the nearest DFT bin frequency $f_k$. The sinusoid can no longer be represented by a single complex exponential at $f_k$, and instead, its energy spreads across multiple bins.

Mathematically, this is due to the finite length of the signal. The DFT implicitly assumes that the signal is periodic with a period equal to the length of the signal $N$. When $f_0$ does not match any $f_k$, the assumption of periodicity introduces discontinuities at the boundaries of the signal, creating artifacts in the frequency domain. These artifacts manifest as energy in frequency bins that are not directly associated with $f_0$, leading to the appearance of spectral leakage.

To illustrate, consider the DFT of the signal $x[n] = A\cos(2\pi\frac{k_0+\delta}{N}n)$, where $k_0$ is an integer, and $\delta$ is a small fractional frequency component such that $f_0 = \frac{(k_0+\delta)f_s}{N}$. The DFT will show a peak not only at $k_0$ but also in adjacent bins $k_0 \pm 1, k_0 \pm 2, \ldots$, depending on the value of $\delta$. The energy spreads over several bins, reducing the sharpness of the spectral peak, which would otherwise be concentrated if $\delta = 0$.

In practice, spectral leakage can be mitigated by techniques such as windowing, where the signal is multiplied by a window function that tapers the signal to zero at the boundaries, reducing the discontinuities and therefore the leakage. However, even with windowing, some degree of leakage is typically unavoidable when $f_0$ does not exactly match a DFT bin.

$\square$

### A.4 Explanation of Proposition 2

The phenomenon of harmonics is a fundamental concept in signal processing, particularly when dealing with signals that exhibit non-linearities or discontinuities. Consider a periodic signal $x(n)$ with a fundamental frequency $f_0$. The fundamental frequency is the inverse of the period $T$ of the signal, i.e., $f_0 = \frac{1}{T}$. For a perfectly sinusoidal signal, the frequency spectrum would ideally consist of a single peak at $f_0$. However, when $x(n)$ contains non-linearities or sharp transitions, these irregularities introduce additional frequency components into the spectrum, known as harmonics. These harmonics are integer multiples of the fundamental frequency, occurring at frequencies $2f_0$ (second harmonic), $3f_0$ (third harmonic), and so on.

Non-linearities in a signal can arise due to various causes, such as amplitude clipping, rectification, or the presence of sharp corners or edges. When a signal with such characteristics is analyzed in the frequency domain, the Fourier transform reveals not only the fundamental frequency but also its harmonics. For example, consider a signal $x(n) = A\cos(2\pi f_0 n)$ that undergoes a non-linear transformation, such as squaring. The resulting signal $y(n) = x^2(n) = A^2\cos^2(2\pi f_0 n)$ can be expressed using the trigonometric identity $\cos^2(\theta) = \frac{1}{2} + \frac{1}{2}\cos(2\theta)$, leading to $y(n) = \frac{A^2}{2} + \frac{A^2}{2}\cos(4\pi f_0 n)$. The frequency spectrum of $y(n)$ now includes a DC component (at $f = 0$) and a component at $2f_0$, the second harmonic.

Discontinuities or sharp transitions in a time series, modeled as abrupt changes in the signal's amplitude, also introduce a wide range of frequency components, particularly higher harmonics. A square wave, for example, alternates between two levels, and its Fourier series expansion reveals that it consists of the fundamental frequency and all odd harmonics (3rd, 5th, 7th, etc.). This behavior is due to the sharp transitions between the high and low states of the square wave.

In practical applications, the presence of harmonics can have significant implications. In audio signals, harmonics can add richness or distortion, depending on whether they are desired or not. In communication systems, harmonics can cause interference, necessitating the use of filters to remove them. In power systems, harmonics can lead to inefficiencies and potential damage to equipment. Therefore, understanding the source of harmonics and how they manifest in a signal's spectrum is crucial for effective signal processing and analysis.

### A.5 Proof of Theorem 2

*Proof.* The parameter count can be divided into two parts: the number of parameters in the SFM and the patch predictor. Given the input sequence length $L$, the number of patches $P$, and the output horizon $H$, the SFM can be viewed as a mapping from $f_c$ frequency components to $f_c$ frequency components (i.e., $\mathbb{C}^{f_c} \to \mathbb{C}^{f_c}$) with each sparse group conducting a linear transformation from $\frac{f_c}{K}$ to $\frac{f_c}{K}$ frequency components. The number of parameters in the SFM can therefore be calculated as:

$$Parameters_{\text{SFM}} = \frac{f_c}{K} \times \frac{f_c}{K} \times K = \frac{f_c^2}{K} \tag{11}$$

The output number of patches can be calculated as $\frac{PH}{L}$, so that the patch predictor can be viewed as a mapping from $\mathbb{C}^P$ to $\mathbb{C}^{\frac{PH}{L}}$, and the number of parameters in the patch predictor can be calculated as:

$$Parameters_{\text{PatchPredictor}} = P \times \frac{PH}{L} = \frac{P^2 H}{L} \tag{12}$$

Therefore, the total number of parameters in FastTF is the sum of the parameters in the SFM and the patch predictor:

$$Parameters = \frac{f_c^2}{K} + \frac{P^2 H}{L} \tag{13}$$

Specifically, after downsampling we get $PM$ subsequences with length $L_{sub} = \frac{L}{PM}$, and then the rFFT operation is applied to each subsequence, yielding another $PM$ subsequences with length $\left\lfloor \frac{L}{2PM} \right\rfloor + 1$. By setting $f_c = \left\lfloor \frac{L}{2PM} \right\rfloor + 1$, the number of parameters in the SFM can be calculated as:

$$Parameters_{\text{SFM}} = \frac{f_c^2}{K} = \frac{\left( \left\lfloor \frac{L}{2PM} \right\rfloor + 1 \right)^2}{K}, \tag{14}$$

and therefore the total number of parameters in FastTF is:

$$Parameters = \frac{\left( \left\lfloor \frac{L}{2PM} \right\rfloor + 1 \right)^2}{K} + \frac{P^2 H}{L}. \tag{15}$$

$\square$

## B  The Reason for not Filtering the Original Time Series

Low-pass filtering before downsampling is essential to prevent spectral aliasing, a phenomenon where higher frequency components of a signal are incorrectly interpreted as lower frequencies. This occurs because downsampling reduces the sampling rate, which can cause the signal's frequency content to exceed the new Nyquist limit (half the new sampling rate). If the original sequence contains frequency components higher than this limit, they can fold back into the lower frequency range, leading to distortion and loss of information. By applying a low-pass filter, these higher frequency components can be effectively removed, ensuring that the downsampled sequence accurately represents the original signal within the new bandwidth. This process preserves the integrity of the signal and prevents aliasing artifacts, which could otherwise compromise the quality of the downsampled data.

However, in FastTF, we do not explicitly filter the original time series before downsampling. This decision is mainly based on the following considerations:

- **Additional Complexity:** Incorporating a low-pass filter before downsampling would introduce additional computational complexity to the model. Specifically, an extra rFFT and irFFT operation would be required to implement the filtering process, increasing the overall computational load. Given that FastTF aims to be a lightweight and efficient model, minimizing unnecessary complexity is crucial to maintain its simplicity and speed.
- **Potential Information Loss:** On the one hand, From a time domain perspective, the single-sequence downsampling process itself acts as a low-pass filter, effectively removing high-frequency components that exceed the new Nyquist limit. This operation dicarded many point

Table 10: Hyperparameters for different datasets and prediction horizons. Here, *cnt* represents the total number of parameters for the configuration. Here *PS* denotes the patch size.

| Dataset | 96 | | | | | 192 | | | | | 336 | | | | | 720 | | | | |
|---|---|---|---|---|---|---|---|---|---|---|---|---|---|---|---|---|---|---|---|---|
| | PS | M | $f_c$ | K | cnt | PS | M | $f_c$ | K | cnt | PS | M | $f_c$ | K | cnt | PS | M | $f_c$ | K | cnt |
| ETTh1 | 48 | 2 | 12 | 2 | **102** | 48 | 2 | 12 | 2 | **132** | 48 | 2 | 12 | 2 | **177** | 48 | 2 | 12 | 2 | **297** |
| ETTh2 | 6 | 1 | 4 | 2 | 1928 | 6 | 1 | 4 | 2 | 3848 | 6 | 1 | 4 | 2 | 6728 | 6 | 1 | 4 | 2 | 14408 |
| Electricity | 4 | 1 | 3 | 1 | 4329 | 4 | 1 | 3 | 1 | 8649 | 4 | 1 | 3 | 1 | 15129 | 4 | 1 | 3 | 1 | 32409 |
| Traffic | 24 | 2 | 6 | 2 | 138 | 24 | 2 | 6 | 2 | 258 | 24 | 2 | 6 | 2 | 438 | 24 | 2 | 6 | 2 | 918 |
| Weather | 12 | 2 | 4 | 2 | 648 | 12 | 2 | 4 | 2 | 1128 | 12 | 2 | 4 | 2 | 1848 | 12 | 2 | 4 | 2 | 3768 |

in the original time series, which will result in severe information loss. However, in FastTF, the downsampling process was repeatedly applied for $M$ times, which actually maintained all the information in the original time series. Therefore, the distortion caused by aliasing can be somehow alleviated. On the other hand, the filter operation before downsampling may also remove some useful information in the original time series, which is not conducive to the subsequent time series forecasting task.

## C  ADDITIONAL EXPERIMENTAL RESULTS

### C.1  TRAINING SETTINGS

In this section, we provide additional details on the training settings used in the experiments. The training process was conducted on a single NVIDIA RTX 4090 GPU with 24GB memory. The optimizer used was Adam with a learning rate of 0.008, a batch size of 256, and a maximum epoch of 100. The model was implemented using PyTorch and trained using the Mean Squared Error (MSE) loss function. The hyperparamter table for each dataset is shown in Table 10. Specifically, the MSE loss was calculated as:

$$\text{MSE} = \frac{1}{N} \sum_{i=1}^{N} (\boldsymbol{y}_i - \hat{\boldsymbol{x}}_i)^2, \tag{16}$$

where $N$ is the number of samples, $\boldsymbol{y}_i$ is the true value (ground truth), and $\hat{\boldsymbol{x}}_i$ is the predicted value.

During Training, we adopt an *early stop* strategy based on the validation loss. Specifically, the training process will be terminated if the validation loss does not decrease for 6 consecutive epochs. The model parameters at the epoch with the lowest validation loss are saved as the final model. Besides, we also adopt the *learning rate decay* strategy. Specifically, the learning rate will be reduced by a factor of 0.6 for every 10 epochs.

### C.2  ABLATION STUDY

The ablation study result is shown in Table 11. The ablation study is conducted on different datasets and prediction horizons to evaluate the impact of different components of FastTF. The *FastTF* denotes the full model, while *FastTF-SFM* and *FastTF-rFFT* represent the models without the SFM and rFFT components, respectively. The results show that transforming the input sequence into the frequency domain using rFFT and applying SFM to mix the frequency information can greatly improve the performance of FastTF across different datasets and prediction horizons.

### C.3  HYPERPARAMETER SEARCHING DETAILS

We searched for the optimal hyperparameters for FastTF on the ETTh1, ETTh2, Traffic, and Weather datasets, and the results are shown in Tables 12, 13, 14, and 15, respectively. It must be noted that, although the results vary across different hyperparamters, the performance of FastTF is generally robust with respect to hyperparameters, as the model consistently achieves competitive results across different settings. This robustness is a key advantage of FastTF, making it easy to deploy in practice without extensive hyperparameter tuning. Specifically, for resource constrained environments,

Table 11: Ablation study results for different datasets and prediction horizons. *FastTF* denotes the full model, while *FastTF-SFM* represents the model without the SFM component.

| Dataset | Ablation | Horizon | | | |
|---|---|---|---|---|---|
| | | 96 | 192 | 336 | 720 |
| ETTh1 | FastTF | **0.350** | **0.388** | **0.419** | **0.416** |
| | FastTF-SFM | 0.396 | 0.425 | 0.448 | 0.439 |
| ETTh2 | FastTF | **0.268** | **0.330** | **0.351** | **0.376** |
| | FastTF-SFM | 0.292 | 0.346 | 0.364 | 0.387 |
| Electricity | FastTF | **0.132** | **0.149** | **0.165** | **0.200** |
| | FastTF-SFM | 0.202 | 0.198 | 0.242 | 0.270 |
| Traffic | FastTF | **0.387** | **0.403** | **0.410** | **0.435** |
| | FastTF-SFM | 0.464 | 0.473 | 0.476 | 0.542 |
| Weather | FastTF | **0.140** | **0.182** | **0.232** | **0.301** |
| | FastTF-SFM | 0.191 | 0.232 | 0.272 | 0.335 |

the model can be deployed with a relatively small number of parameters while maintaining strong forecasting performance.

Table 12: Parameter search results for ETTh1. The number in the paratheis represents the corresponding parameter count. Here the patch size $PS$ is set to 48.

| Horizon | 96 | | | | 720 | | | |
|---|---|---|---|---|---|---|---|---|
| M \ K | 1 | 2 | 3 | 6 | 1 | 2 | 3 | 6 |
| 1 | 0.350(655) | 0.350(318) | 0.352(222) | 0.351(126) | 0.418(850) | 0.416(513) | 0.417(417) | 0.416(321) |
| 2 | 0.351(199) | **0.350**(102) | 0.353(78) | 0.355(54) | 0.416(394) | **0.416**(297) | 0.420(273) | 0.419(249) |
| 4 | 0.353(79) | 0.355(48) | 0.369(42) | – | 0.421(274) | 0.420(243) | 0.433(237) | – |
| 8 | 0.359(46) | 0.362(38) | – | – | 0.425(241) | 0.417(233) | – | – |

Table 13: Parameter search results for ETTh2. Here the patch size $PS$ is set to 6.

| Horizon | 96 | | | | 720 | | | |
|---|---|---|---|---|---|---|---|---|
| M \ K | 1 | 2 | 3 | 6 | 1 | 2 | 3 | 6 |
| 1 | 0.268(1936) | **0.268**(1928) | – | – | 0.376(14416) | **0.376**(14408) | – | – |
| 2 | 0.270(1924) | – | – | – | 0.378(14404) | – | – | – |
| 4 | – | – | – | – | – | – | – | – |
| 8 | – | – | – | – | – | – | – | – |

## C.4 RESULTS ON MORE DATASETS

In this section we provide additional experimental results on *ETTm1*, *ETTm2*, and *Exchange* datasets. The introduction of these datasets can be found in Section E, and the results are summarized in Table 17.

## C.5 CRITICAL DIFFERENCE DIAGRAM

The critical difference diagram is a statistical tool used to compare multiple methods across different datasets and horizons. It is based on the Friedman test, a non-parametric statistical test that determines whether there are significant differences between the methods' performance. The critical difference diagram visualizes the average ranks of the methods and indicates which methods are significantly different from each other based on the critical difference value. The critical difference value is calculated using the Nemenyi test, which takes into account the number of datasets and methods being compared. The result is shown in Figure 10.

Table 14: Parameter search results for Traffic. Here the patch size $PS$ is set to 24.

| Horizon | 96 | | | | 720 | | | |
|---|---|---|---|---|---|---|---|---|
| M \ K | 1 | 2 | 3 | 6 | 1 | 2 | 3 | 6 |
| 1 | 0.390(289) | 0.388(192) | 0.390(168) | 0.389(156) | 0.447(1069) | 0.446(972) | 0.448(948) | 0.456(924) |
| 2 | 0.389(169) | **0.387**(138) | 0.393(132) | – | 0.447(949) | **0.446**(918) | 0.449(912) | – |
| 4 | 0.391(136) | 0.393(128) | – | – | 0.451(916) | 0.452(908) | – | – |
| 8 | 0.396(124) | – | – | – | 0.455(904) | – | – | – |

Table 15: Parameter search results for Weather. Here the patch size $PS$ is set to 12.

| Horizon | 96 | | | | 720 | | | |
|---|---|---|---|---|---|---|---|---|
| M \ K | 1 | 2 | 3 | 6 | 1 | 2 | 3 | 6 |
| 1 | 0.142(529) | 0.141(498) | 0.143(492) | – | 0.302(3649) | 0.301(3618) | 0.306(3612) | – |
| 2 | 0.142(496) | **0.140**(488) | – | – | 0.303(3616) | **0.301**(3608) | – | – |
| 4 | 0.145(484) | – | – | – | 0.306(3604) | – | – | – |
| 8 | – | – | – | – | – | – | – | – |

### C.6 ADDITIONAL PREDICTION RESULTS

Additional prediction results in the training set are shown in Figures 11, 12, 13, 14, 15, 16, 17, 18, 19, 20. The results show that FastTF can effectively capture the underlying patterns in the time series data and make accurate predictions across different datasets and prediction horizons.

## D THE CODE BUG

In Dec. 2023, an anonymous researcher pointed out a long existing bug in the source code of a series of time series forecasting models. The bug can be traced back to the implementation of Informer (Zhou et al., 2021), and has already affected a series of subsequent works, including DLinear, Autoformer, Fedformer, PatchTST, Koopa, FITS, and TimeMixer, etc. The bug is related to the calculation of the settings of the test dataloader, **where the *drop_last* parameter is incorrectly set to *True* by default.** This setting causes the last incomplete batch of the test dataloader to be dropped, leading to incorrect evaluation results. Empirically, the bug significantly improved the performance of the models on the ETT datasets, and the impact on other datasets is relatively marginal (Qiu et al., 2024). The bug has been fixed in the latest version of the source code, and the corrected results are presented in this paper.

## E DETAIL OF THE PUBLIC DATASETS

1. **Weather**: This dataset contains 21 meteorological indicators such as humidity and air temperature for the year 2020 in Germany.

2. **Traffic**: Contains road occupancy rates measured by 862 different sensors across San Francisco Bay Area freeways over a span of two years. The data is sourced from the California Department of Transportation.

3. **Electricity**: Comprises hourly electricity consumption data of 321 clients, recorded between 2012 and 2014.

4. **Exchange**: Includes daily exchange rates of eight different countries from 1990 to 2016. patient data in the United States from 2002 to 2021. The dataset includes seven indicators such as the number of ILI patients across different age groups and the ratio of ILI patients to the total number of patients. Data is provided by the Centers for Disease Control and Prevention (CDC).

5. **ETT (Electricity Transformer Temperature)**: Contains data collected from electricity transformers using seven sensors, capturing variables such as load, oil temperature, etc. The dataset is split into two sub-datasets labeled as 1 and 2, corresponding to two different electric transformers from separate counties in China. Each sub-dataset includes two different time resolutions: 15 minutes and 1 hour, denoted as *m* and *h* respectively. Thus, there are four ETT datasets: ETTh1, ETTh2, ETTm1, and ETTm2.

Table 16: Complete form of statistical information for the datasets used in the experiments.

| Dataset | Weather | Traffic | Exchange | Electricity | ETTh1 | ETTh2 | ETTm1 | ETTm2 |
|---|---|---|---|---|---|---|---|---|
| Dataset Size | 52696 | 17544 | 7207 | 26304 | 17420 | 17420 | 69680 | 69680 |
| Variable Number | 21 | 862 | 8 | 321 | 7 | 7 | 7 | 7 |
| Sampling Frequency | 10 mins | 1 hour | 1 day | 1 hour | 1 hour | 1 hour | 15 mins | 15 mins |

Table 17: **Comparison of different methods across various datasets and horizons.** The best 3 results are highlighted **red**, blue, *green*, respectively.

| Dataset | ETTm1 | | | | ETTm2 | | | | Exchange | | |
|---|---|---|---|---|---|---|---|---|---|---|---|
| Horizon | 96 | 192 | 336 | 720 | 96 | 192 | 336 | 720 | 96 | 192 | 336 |
| FEDformer (2022b) | 0.326 | 0.365 | 0.392 | 0.446 | 0.180 | 0.252 | 0.324 | 0.410 | 0.139 | 0.256 | 0.426 |
| TimesNet (2023) | 0.338 | 0.371 | 0.410 | 0.478 | 0.187 | 0.249 | 0.321 | 0.497 | 0.107 | 0.226 | 0.367 |
| PatchTST (2023) | **0.290** | **0.332** | **0.366** | 0.416 | 0.165 | 0.220 | 0.274 | 0.362 | 0.093 | 0.192 | 0.350 |
| DLinear (2023) | *0.299* | *0.335* | 0.369 | *0.425* | *0.167* | *0.224* | *0.281* | 0.397 | 0.081 | **0.157** | *0.305* |
| U-Mixer(2024) | 0.317 | 0.369 | 0.395 | 0.443 | 0.178 | 0.243 | 0.331 | 0.434 | 0.087 | *0.171* | 0.285 |
| Koopa (2024) | 0.294 | 0.337 | 0.380 | 0.426 | 0.171 | 0.226 | 0.283 | 0.394 | *0.083* | 0.184 | 0.331 |
| MICN (2023) | 0.314 | 0.359 | 0.398 | 0.459 | 0.178 | 0.245 | 0.295 | *0.389* | 0.102 | 0.172 | **0.272** |
| **FastTF (ours)** | 0.302(4th) | 0.334 | *0.372* | **0.415** | **0.162** | **0.215** | **0.266** | **0.349** | **0.080** | 0.167 | 0.304 |

# F   THE INDIVIDUAL CONFIGURATION

In practical applications, time series datasets are often multi-channel. Let $C$ denote the number of input channels, then a look-back window of data can be represented as $\boldsymbol{X} \in \mathbb{R}^{C \times L}$. For channel-independent models like FITS and DLinear, there are generally two training strategies:

1. Train a separate single-channel model for each channel, with independent weights across channels, referred to as the *Individual* configuration.

2. Use a shared set of weights across all channels.

For FastTF, we also employ the *Individual* configuration on the weather dataset, but with a key difference: we train a SFM for each channel independently, while the patch predictor shares weights across channels. The motivation behind this strategy is to reduce the model's parameter count and mitigate overfitting. Specifically, the SFM integrates frequency information over short time intervals, where different channels exhibit varying short-term behavior in the weather dataset. Conversely, the patch predictor captures long-term trend changes across patches, which tend to be consistent across different channels.

# G   PARAMETER TABLE FOR FASTTF

The detailed parameter table of patch predictor and SFM in FastTF is shown in Table 18 and Table 19, respectively. The parameter count of the patch predictor is calculated based on the patch size, prediction horizon, and look-back window, while the parameter count of the SFM is determined by the patch size, downsampling factor, and sparse group number.

Table 18: The parameter count of **patch predictor** in FastTF with different patch sizes, prediction horizons and look-back windows.

| patch size | $L/P = 12$ | | | | | $L/P = 24$ | | | | | $L/P = 48$ | | | | |
|---|---|---|---|---|---|---|---|---|---|---|---|---|---|---|---|
| Horizon \ Look-back | 96 | 192 | 336 | 672 | 720 | 96 | 192 | 336 | 672 | 720 | 96 | 192 | 336 | 672 | 720 |
| 96 | 64 | 128 | 224 | 448 | 480 | 16 | 32 | 56 | 112 | 120 | 4 | 8 | 14 | 28 | 30 |
| 192 | 128 | 256 | 448 | 896 | 960 | 32 | 64 | 112 | 224 | 240 | 8 | 16 | 28 | 56 | 60 |
| 336 | 224 | 448 | 784 | 1568 | 1680 | 56 | 112 | 196 | 392 | 420 | 14 | 28 | 49 | 98 | 105 |
| 720 | 480 | 960 | 1680 | 3360 | 3600 | 120 | 240 | 420 | 840 | 900 | 30 | 60 | 105 | 210 | 225 |

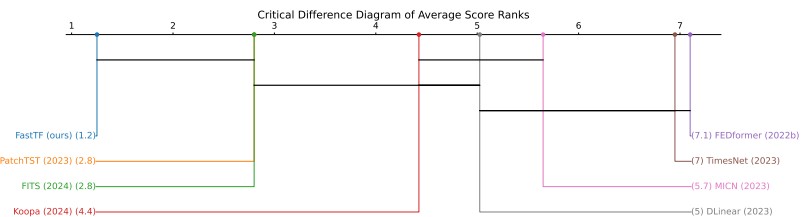

Figure 10: Critical difference diagram for the comparison of different methods across various datasets and horizons. The methods are ranked based on the average rank across all datasets and horizons.

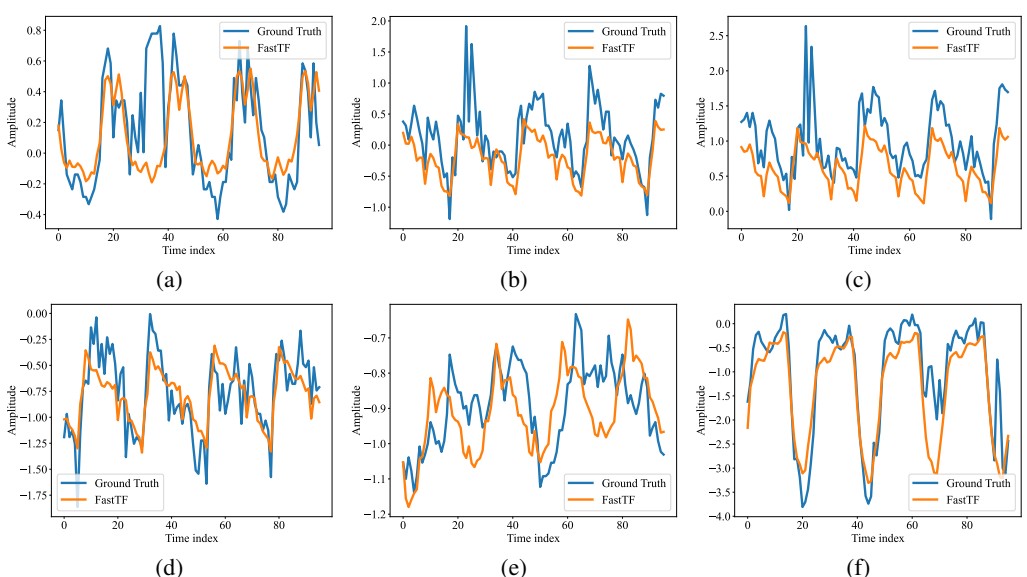

Figure 11: Additional prediction results on the ETTh1 dataset with $H = 96$.

# H DETAILED COMPLEXITY ANALYSIS

In main text, we focus on calculating the parameter count of FastTF, which is a key indicator of model complexity. However, the computational complexity of FastTF is also an important aspect to consider, as it directly affects the model's training and inference efficiency. Here, we provide a detailed analysis of the computational complexity of FastTF, focusing on the forward pass of the model.

**Lemma 1.** *One complex multiplication requires at least 3 real multiplications.*

*Proof.* for $z_1 = a + bj$, $z_2 = c + dj$, $z = z_1 + z_2$, let $p = (a + b)(c - d), q = ac, r = bd$, then the real part of $z$ is $q - r$, and the imaginary part of $z$ is $p - q + r$. To calculate $p, q$ and $r$, we need 3 real multiplications, therefore one complex multiplication requires at least three real multiplications. □

**Theorem 3** (The number of multiplications in FastTF). *Given the input sequence length $L$, the number of patches $P$, the downsampling factor $M$, the cut-off frequency $f_c$, the number of sparse groups $K$, the number of multiplication operations in N-point rFFT (denoted as $M_{rFFT}^N$) and N-point irFFT (denoted as $M_{irFFT}^N$), the total number of multiplication operations in FastTF (to process a single channel input sequence $\boldsymbol{x} \in \mathbb{R}^L$) can be calculated as:*

$$Mul = PM \times M_{rFFT}^{L/PM} + \frac{MHP}{L} \times M_{irFFT}^{L/PM} + PMf_c^2/K + f_cMP^2H/L \qquad (17)$$

*Specifically, the number of real-valued multiplication operations is 3 times the number of complex-valued multiplication operations, so the total number of real-valued multiplication operations in*

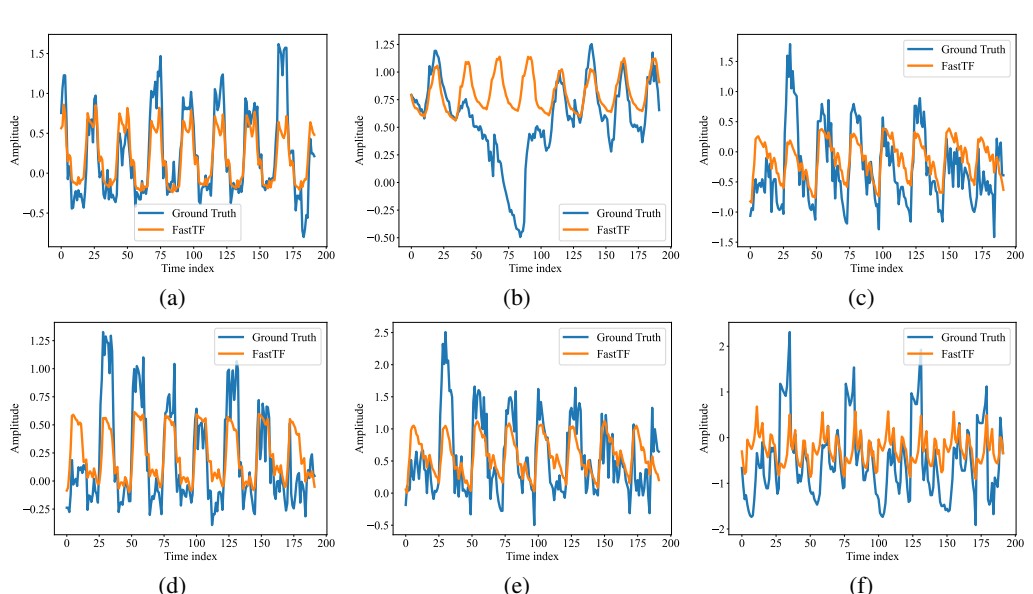

Figure 12: Additional prediction results on the ETTh1 dataset with $H = 192$.

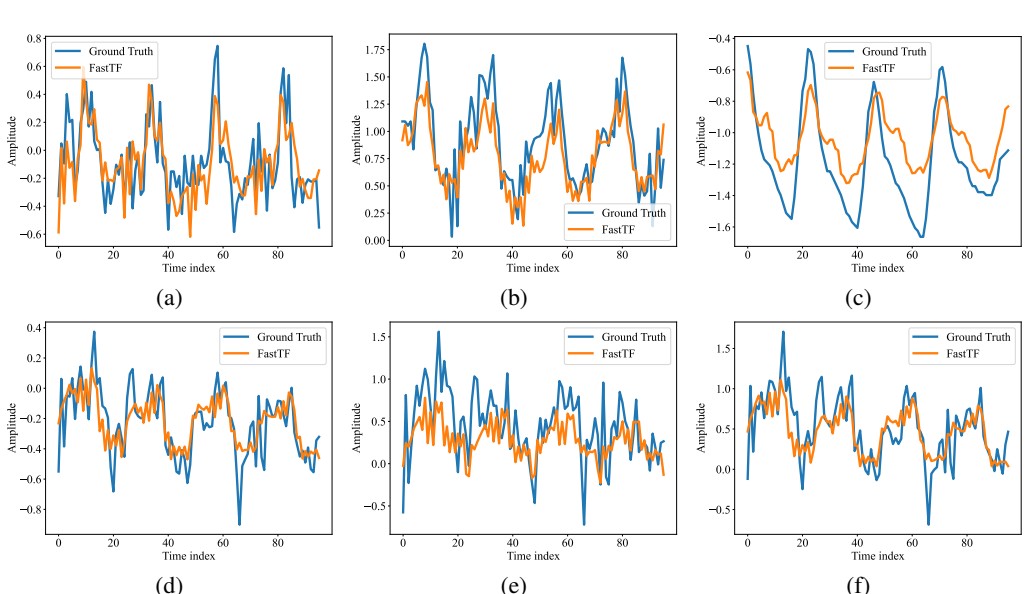

Figure 13: Additional prediction results on the ETTh2 dataset with $H = 96$.

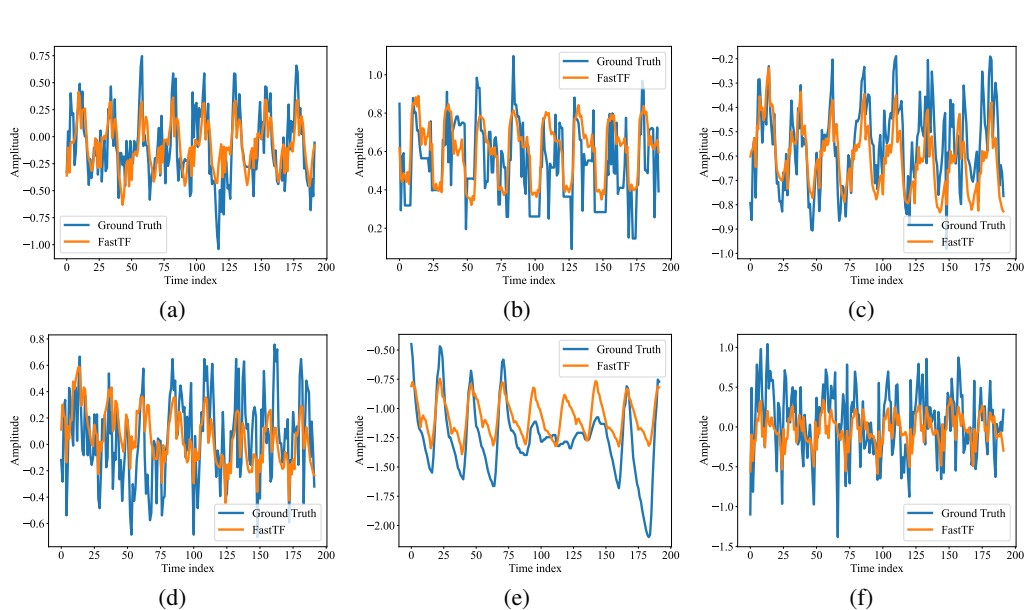

Figure 14: Additional prediction results on the ETTh2 dataset with $H = 192$.

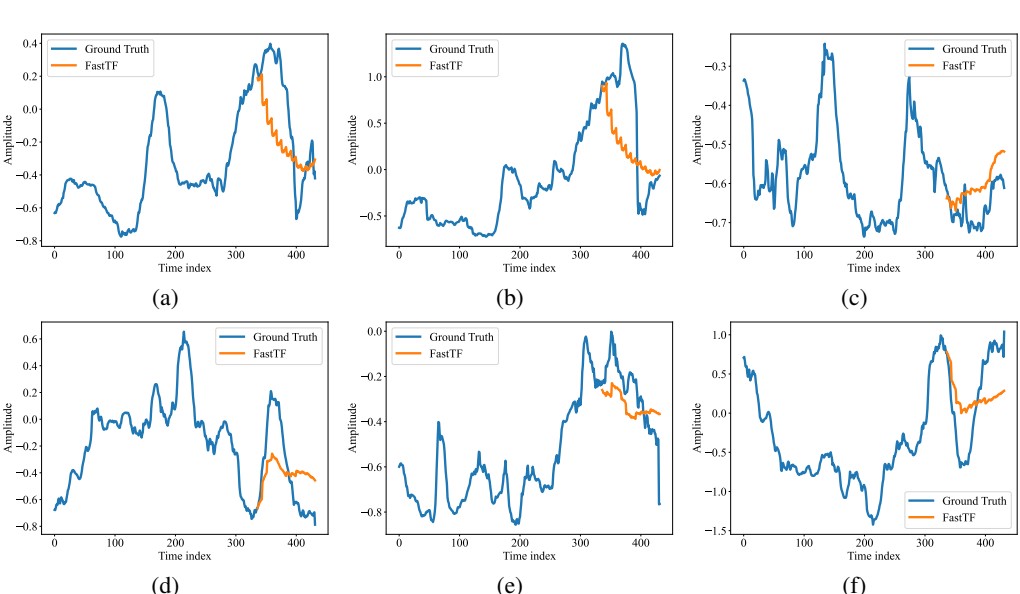

Figure 15: Additional prediction results on the weather dataset with $H = 96$.

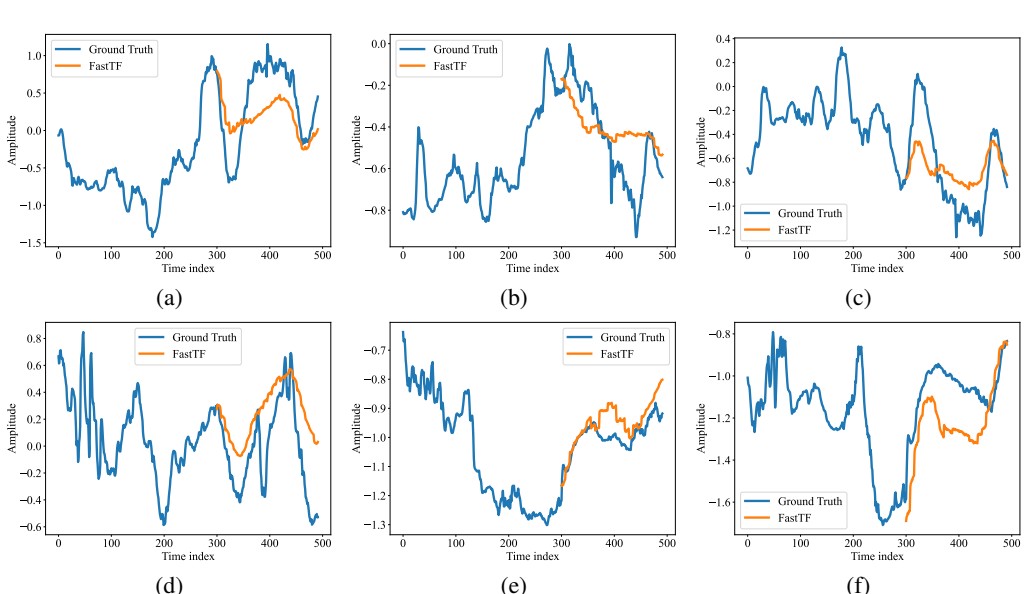

Figure 16: Additional prediction results on the weather dataset with $H = 192$.

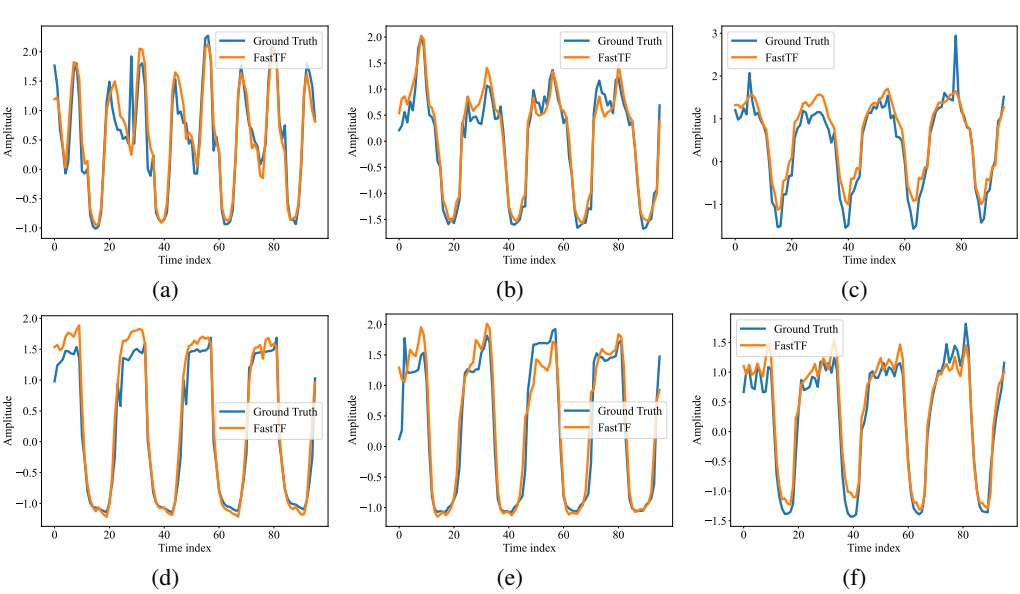

Figure 17: Additional prediction results on the Electricity dataset with $H = 96$.

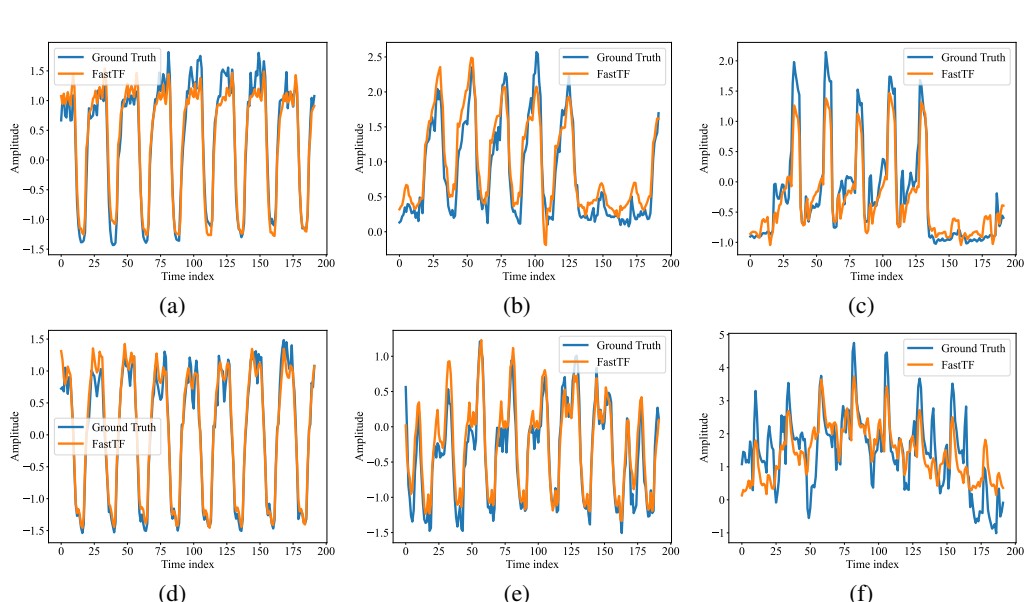

Figure 18: Additional prediction results on the Electricity dataset with $H = 192$.

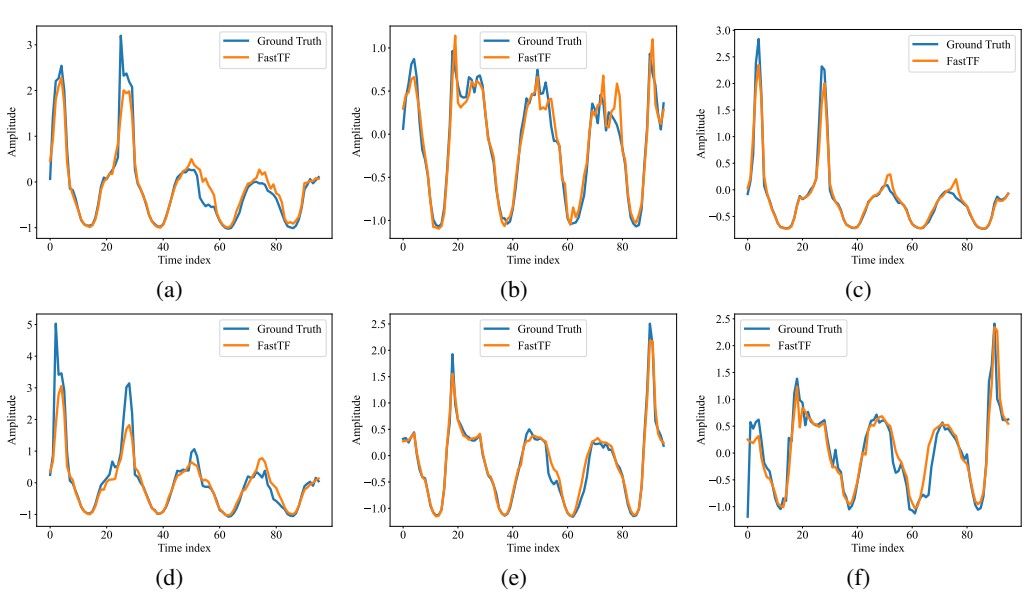

Figure 19: Additional prediction results on the Traffic dataset with $H = 96$.

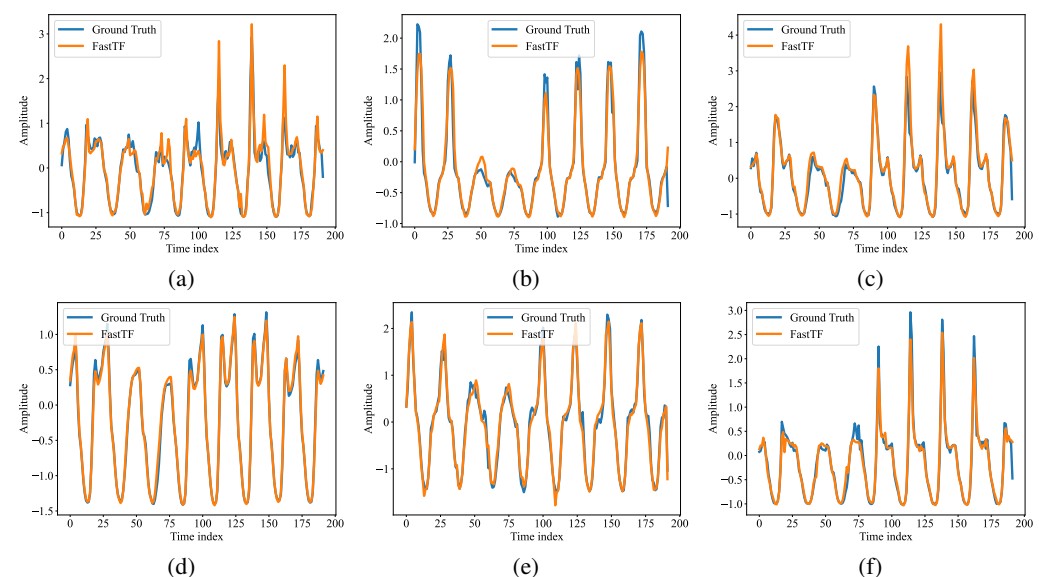

Figure 20: Additional prediction results on the Traffic dataset with $H = 192$

Table 19: The parameter count of **SFM** in FastTF with different patch sizes, downsampling factors and sparse group numbers. Note that the values in parentheses represent the corresponding cut-off frequency.

| patch size | $L/P = 12$ | | | | $L/P = 24$ | | | | | | $L/P = 48$ | | | | | |
|---|---|---|---|---|---|---|---|---|---|---|---|---|---|---|---|---|
| $K$ \ $M$ | 1 | 2 | 4 | 6 | 1 | 2 | 4 | 6 | 8 | 12 | 1 | 2 | 4 | 8 | 12 | 24 |
| 1 | 49(7) | 6(4) | 4(2) | 4(2) | 169(13) | 49(7) | 16(4) | 9(3) | 4(2) | 4(2) | 625(25) | 169(13) | 49(7) | 16(4) | 9(3) | 4(2) |
| 2 | 18(6) | 8(4) | – | – | 72(12) | 18(6) | 8(4) | – | – | – | 288(24) | 72(12) | 18(6) | 8(4) | – | – |
| 3 | 12(6) | – | – | – | 48(12) | 12(6) | – | – | – | – | 192(24) | 48(12) | 12(6) | – | – | – |
| 4 | – | – | – | – | 36(12) | – | – | – | – | – | 144(24) | 36(12) | – | – | – | – |
| 6 | – | – | – | – | 24(12) | – | – | – | – | – | 96(24) | 24(2) | – | – | – | – |

*FastTF can be calculated as:*

$$Mul_{real} = 3 \times Mul$$

$$= 3(PMM_{rFFT}^{L/PM} + \frac{MHP}{L}M_{irFFT}^{L/PM} + PMf_c^2/K + f_cMP^2H/L) \tag{18}$$

*If we omit the rFFT and irFFT operations and only consider the multiplication operations in the neural network part, the number of multiplication operations in FastTF can be simplified as:*

$$Mul_{nn,real} = 3PMf_c^2/K(SFM) + 3f_cMP^2H/L(Patch\ Predictor) \tag{19}$$

*Proof.* The number of multiplication operations in the rFFT and irFFT operations can be calculated based on the input sequence length $L$, the downsampling factor $M$, and the cut-off frequency $f_c$. Specifically, the rFFT operation is applied to $PM$ subsequences with length $L_{sub} = \frac{L}{PM}$, and the irFFT operation is applied to $PM$ subsequences. The number of multiplication operations in the rFFT and irFFT operations can be calculated as $PM \times M_{rFFT}^{L/PM}$ and $\frac{MHP}{L} \times M_{irFFT}^{L/PM}$, respectively.

The number of multiplication operations in the SFM and patch predictor can be calculated based on the number of sparse groups $K$, the cut-off frequency $f_c$, the number of patches $P$, and the prediction horizon $H$. Specifically, the SFM applies $\mathbb{C}^{f_c} \to \mathbb{C}^{f_c}$ transformations to each subsequence, results in $Mf_c^2/K$ complex-valued multiplication operations. And similarly, the number of multiplication operations in the patch predictor is $f_cMP^2H/L$. The total number of multiplication operations in FastTF is the sum of the multiplication operations in the rFFT, irFFT, SFM, and patch predictor, which gives the formula for Mul.

The number of real-valued multiplication operations is 3 times the number of complex-valued multiplication operations, as each complex multiplication operation can be decomposed into 4 real-valued

Table 20: The number of multiplication operations in FastTF, DLinear, and FITS with different prediction horizons and look-back windows. The kernel size in DLinear is set to 25, the cut-off frequency for FITS is set to half of the rFFT sequence length, the patch size is set to 48, the $f_c$ is set to 12, the downsampling factor $M$ is set to 2, and the sparse group number $K$ is set to 3.

| patch size Look-back Horizon | DLinear | | | | | FITS | | | | | FastTF | | | | |
|---|---|---|---|---|---|---|---|---|---|---|---|---|---|---|---|
| | 96 | 192 | 336 | 672 | 720 | 96 | 192 | 336 | 672 | 720 | 96 | 192 | 336 | 672 | 720 |
| 96 | 20.8K | 41.7K | 72.9K | 145.8K | 156.2K | 3.5K | 10.4K | 27.2K | 96.8K | 110.2K | 864 | 1728 | 3024 | 6048 | 6480 |
| 192 | 39.3K | 78.5K | 137.4K | 274.8K | 294.5K | 5.2K | 13.8K | 33.3K | 108.9K | 123.1K | 1152 | 2304 | 4032 | 8064 | 8640 |
| 336 | 66.9K | 133.8K | 234.2K | 468.4K | 501.8K | 7.8K | 19.0K | 42.3K | 127.0K | 142.6K | 1584 | 3168 | 5544 | 11088 | 11880 |
| 720 | 140.6K | 281.3K | 492.2K | 984.5K | 1054.8K | 14.7K | 32.9K | 66.5K | 175.4K | 194.4K | 2736 | 5472 | 9576 | 19152 | 20520 |

multiplication operations. Therefore, the total number of real-valued multiplication operations in FastTF is 3 times Mul, which gives the formula for $\text{Mul}_{\text{real}}$.

If we omit the rFFT and irFFT operations and only consider the multiplication operations in the neural network part, the number of multiplication operations in FastTF can be simplified as the sum of the multiplication operations in the SFM and patch predictor, which gives the formula for $\text{Mul}_{\text{nn,real}}$.  □

**Theorem 4** (The number of multiplications in DLinear and FITS). *We only consider the multiplication operations in the neural network part, so that given the input sequence length $L$, the prediction horizon $H$, and the cut-off frequency $f_c^{FITS}$, the length ratio $\eta$ and the kernel size for the average pooling $K_{avg}$, the number of multiplication operations in DLinear can be calculated as:*

$$Mul_{DLinear} = 2LH + K_{avg}L \tag{20}$$

*The number of multiplication operations in FITS can be calculated as:*

$$Mul_{FITS,nn,real} = 3f_c^{FITS} \lfloor f_c^{FITS}\eta \rfloor \tag{21}$$

*Proof.* The proof is trivial and omitted here.  □

The comparison of the number of multiplication operations in FastTF, DLinear, and FITS is shown in Table 20. The results show that FastTF has a significantly lower number of multiplication operations compared to DLinear and FITS. Specifically, for 720-720 prediction horizon and look-back window, FastTF reached only around 2% and 10% of the multiplication operations in DLinear and FITS, respectively.

# I  TRAINING ACCELERATION

To further reduce the training time of FastTF, a preprocessing step can be applied to the input data to accelerate the training process. Specifically, the input data can be downsampled and transformed using the rFFT operation in advance, and the resulting data can be stored for training. This preprocessing step can significantly reduce the computational cost of the rFFT operation during training, as the rFFT operation is computationally intensive and can be a bottleneck in the training process. The detailed algorithm for preprocessing and training with FastTF is shown in Algorithm 1.

# J  DEVELOPMENT ON FPGA CHIPS

## J.1  BRIEF INTRODUCTION TO FPGA CHIPS

FPGA, or Field-Programmable Gate Array, is a type of integrated circuit that allows users to configure hardware functionality after manufacturing. This versatility enables the implementation of custom digital circuits tailored to specific applications, making FPGAs ideal for tasks such as signal processing, data processing, and system control. FPGAs feature a matrix of programmable logic blocks, interconnections, and input/output pins, allowing for high parallel processing capabilities and real-time operation. They are widely used in various industries, including telecommunications, automotive, and aerospace, due to their low latency, low power consumption, and adaptability to changing requirements.

---

**Algorithm 1** Preprocess and Train with FastTF

---

**Require:** Training data $\boldsymbol{X} = \{\boldsymbol{x}^{(1)}, \boldsymbol{x}^{(2)}, \ldots, \boldsymbol{x}^{(N)}\}$, Downsample factor $M$, cut-off frequency $f_c$, Batch size $B$
**Ensure:** Trained model $\theta$
 1: **Preprocessing:**
 2: **for** $i = 1$ to $N$ **do**
 3:   $\boldsymbol{x}_{\text{downsampled}}^{(i)} \leftarrow \text{Downsample}(\boldsymbol{x}^{(i)}, M)$
 4:   $\boldsymbol{X}_{\text{rFFT}}^{(i)} \leftarrow \text{rFFT}(\boldsymbol{x}_{\text{downsampled}}^{(i)})$
 5:   $\boldsymbol{X}_{\text{preprocessed}}^{(i)} \leftarrow \text{Cut}(\boldsymbol{X}_{\text{rFFT}}^{(i)}, f_c)$
 6: **end for**
 7: Store preprocessed data $\boldsymbol{X}_{\text{preprocessed}}$
 8: **Training:**
 9: **for** each training epoch **do**
10:   **for** each mini-batch of size $B$ **do**
11:     Sample a batch $\boldsymbol{X}_{\text{batch}} \subset \boldsymbol{X}_{\text{preprocessed}}$
12:     Forward pass: Predict $\hat{\boldsymbol{X}} \leftarrow \text{FastTF}(\boldsymbol{X}_{\text{batch}})$
13:     Compute loss $L(\hat{\boldsymbol{X}}, \boldsymbol{Y})$
14:     Backpropagate gradients
15:     Update model parameters $\theta$
16:   **end for**
17: **end for**

---

## J.2 ZYNQ ULTRASCALE+ RFSOC ZCU208 EVALUATION KIT

The Zynq™ UltraScale+™ RFSoC ZCU208 Evaluation Kit is an ideal platform for out-of-the-box RF evaluation and cutting-edge application development. It features the Zynq UltraScale+ RFSoC ZU48DR, which integrates eight 14-bit 5GSPS ADCs, eight 14-bit 10GSPS DACs, and eight soft-decision forward error correction (SD-FEC) cores, making it suitable for RF-class applications. The key features of the ZCU208 Evaluation Kit are shown in Table 21.

Table 21: Key Parameters of Zynq UltraScale+ RFSoC ZCU208

| Parameter | Value |
|---|---|
| 14-bit, 5.0 GSPS RF-ADC Count | 8 |
| 14-bit, 10.0 GSPS RF-DAC Count | 8 |
| SD-FEC Cores | 8 |
| System Logic Cells (K) | 930 |
| Memory (Mb) | 60.5 |
| DSP Slices | 4272 |
| 33G Transceivers | 16 |
| Maximum I/O Pins | 347 |

The board features are shown in Figure 21.

## J.3 USAGE DETAILS

The resource utilization of FastTF on the ZCU208 Evaluation Kit is shown in Figure 22. We now give a brief introduction for clock, BRAM, and DSP, shown below:

- **Clock** The clock in an FPGA provides the timing signal that synchronizes all operations within the device. It controls the flow of data by determining when actions such as data processing or memory reads/writes occur. Multiple clock domains can exist in an FPGA design to drive different parts of the logic at different frequencies, which helps optimize performance and reduce power consumption.

- **BRAM (Block RAM)** BRAM (Block Random Access Memory) is a dedicated on-chip memory resource available within the FPGA. It provides high-speed memory that can be used for data storage, buffering, or caches. BRAM blocks are widely used in applications requiring temporary

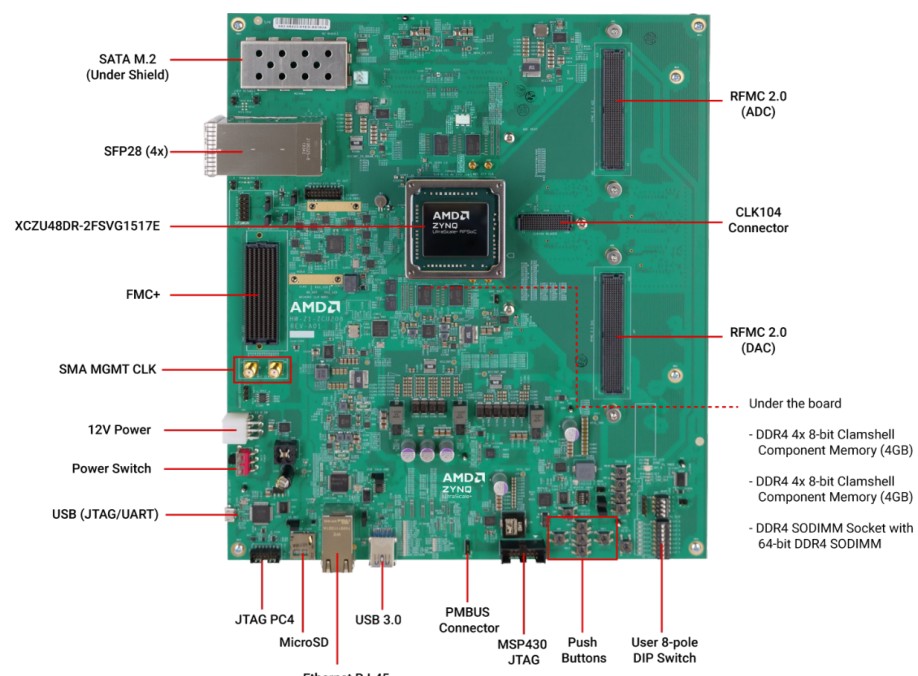

Figure 21: Zynq UltraScale+ RFSoC ZCU208 Evaluation Kit

data storage, such as signal processing, image processing, and communication systems. Unlike external memory, BRAM is tightly integrated into the FPGA fabric, making data access faster and more efficient.

- **DSP (Digital Signal Processing) Slices** DSP slices are specialized hardware units in an FPGA that are optimized for performing arithmetic operations such as multiplication, addition, and accumulation, which are common in digital signal processing tasks. These slices allow FPGAs to efficiently handle operations like filtering, fast Fourier transforms (FFT), and other real-time processing tasks. By leveraging DSP slices, designers can offload critical signal processing operations from general-purpose logic, improving performance and reducing resource usage.

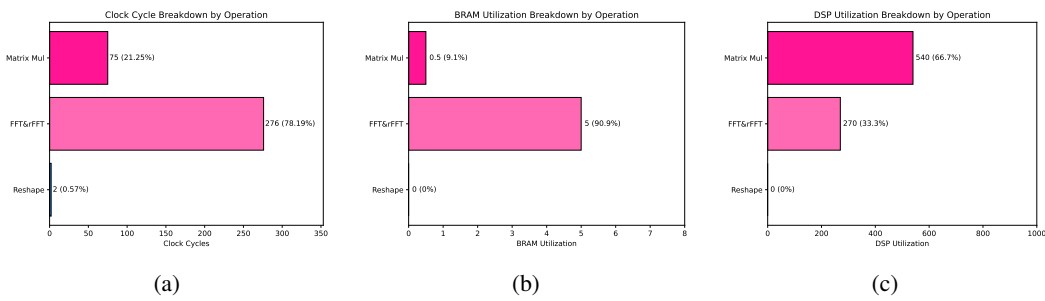

| (a) | (b) | (c) |

Figure 22: Resource utilization of FastTF on ZCU208 Evaluation Kit. (a) Break down by clock cycles (b) Break down by BRAM utilization. (c) Break down by DSP utilization.

Please note that the results may vary depending on the specific FPGA chip and the implementation details, for example, the algorithm for FFT.

## J.4 FUTURE WORK

In this paper, we only present a hardware implementation scheme. Future work can be further explored in the following aspects:

- **A more streamlined hardware implementation of FastTF** Theoretically, FastTF requires significantly less computation than other algorithms, which presents opportunities for reducing the computational resources required on the FPGA.

- **Integration with other hardware accelerators:** Implementation on additional hardware platforms: In the near future, we aim to deploy this algorithm on a range of other hardware devices, including embedded systems such as Raspberry Pi, ESP32, and STM32, to fully verify the performance of the proposed algorithm.

## K    CORE CODE FOR FASTTF

We put part of the core code of FastTF in List 1. The complete code will be released upon acceptance. The code consists the forwawrd pass of FastTF, which is based on the PyTorch framework. However, the definition of the model architecture and the training process are not included in the code snippet.

```python
def forward(self, x):
    batch_size = x.shape[0]
    seq_mean = torch.mean(x, dim=1).unsqueeze(1)
    x = (x - seq_mean).permute(0, 2, 1)

    x = x.reshape(-1, self.seg_num_x, self.num_sampling, self.
        down_sampling).permute(0, 1, 3, 2) # bc,n,period,samp
    x = torch.fft.rfft(x, dim=3)[:, :, :, :self.cut_freq]
    x = x.reshape(-1, self.enc_in, self.seg_num_x, self.down_sampling,
        self.cut_freq)

    if self.flinear_individual:
        x = x.reshape(batch_size, self.enc_in, self.seg_num_x, self.
            down_sampling, self.flinear_sparse_num, self.in_sparse_freq)
        # print(x.shape)
        x = torch.einsum('bcsdft,cfet->bcsdfe', x, self.flinear_weight)
            + x
        # print(y.shape)
    else:
        x = x.reshape(batch_size, self.enc_in, self.seg_num_x, self.
            down_sampling, self.flinear_sparse_num, self.in_sparse_freq)
        x = torch.einsum('bcsdft,fet->bcsdfe', x, self.flinear_weight) +
             x

    x = x.reshape(batch_size, self.enc_in, self.seg_num_x, self.
        down_sampling, self.cut_freq)
    x = x.permute(0, 1, 3, 4, 2)  # b,c,period,samp,n

    if self.linear_individual:
        x = x.reshape(batch_size, self.enc_in, self.down_sampling, self.
            group, self.in_group_freq, self.seg_num_x)
        tmp = torch.einsum('bcfgkn,cgyn->bcfgky', x, self.linear_weight)
    else:

        x = x.reshape(batch_size, self.enc_in, self.down_sampling, self.
            group, self.in_group_freq, self.seg_num_x)
        tmp = torch.einsum('bcfgkn,gyn->bcfgky', x, self.linear_weight)

    x = tmp.reshape(batch_size, self.enc_in, self.down_sampling, self.
        cut_freq, self.seg_num_y)
    tmp2 = torch.zeros([x.size(0), x.size(1), x.size(2), self.
        num_sampling // 2 + 1, x.size(4)], dtype=x.dtype).to(x.device)
    tmp2[:, :, :, :self.cut_freq, :] = x

    y = tmp2.permute(0, 1, 4, 2, 3)

    y = torch.fft.irfft(y, dim=4).permute(0, 1, 2, 4, 3)

    y = y.reshape(batch_size, self.enc_in, self.pred_len)

    y = y.permute(0, 2, 1) + seq_mean

    return y
```

Listing 1: Example Python Code

