# OpenReview forum: "FastTF: 4 Parameters are All You Need for Long-term Time Series Forecasting"
_ICLR.cc/2025/Conference — Submitted to ICLR 2025_

### Official Review · Reviewer_4ASm · 2024-10-31

**Soundness:** 3
**Presentation:** 2
**Contribution:** 2
**Rating:** 5
**Confidence:** 4

**Summary:**

The authors introduce FastTF, a model architecture for time-series forecasting, with the goal of being light-weight while maintaining a competitive performance. FastTF uses two layers of patching in time, then an rFFT on each subpatch, truncated after a chosen cutoff frequency. The learnable parameters of the network are in a blockwise diagonal linear layer for the frequency space subpatches, mixing information within each patch, and another linear layer that mixes information between patches, with the weights being shared across one of the patching dimensions. Afterwards, the frequency data is padded, the FFT inverted and the time data reshaped to obtain the model output. Through this use of sparsity and weight sharing the total number of parameters of this model architecture is significantly lower than in other approaches for time-series forecasting. The authors show competitive performance of FastTF for prediction tasks on several standard time-series datasets and prediction horizons and present a study on the impact of different hyperparameter choices in FastTF as well as a small studies on generalizability, converge speed, and deployability on an FPGA.

**Strengths:**

- Overall the authors fulfilled their aim: FastTF seems to be very light-weight with competitive performance
- Fairly extensive hyperparameter study
- Architectural choices are motivated by and take advantage of empirical observations of weight matrix structures, i.e. weight matrix sparseness as the motivation for the SFM
- The authors also present prediction results that do not perform well, e.g. figures 15 and 16 in the appendix.

**Weaknesses:**

- (major) The case study in Section 5.4 shows that ETTh1 can be predicted with high accuracy from local means; this can be done within FastTF, but is not a good example of its strengths since neither the full expressivity of the Fourier representation nor the SFM is used; the Fourier transforms just add unnecessary overhead here; in summary, this is a finding about the dataset, not the FastTF architecture and thus should not be in the main text.
- (major) The patch size $P$ is always an integral multiple of the fundamental frequency, e.g. 24h for ETTh. For Transformer architectures it has been shown that patching the data like this can improve the performance significantly (see also below the remark about related work), so it cannot be excluded that this (and not the specific structure of the architecture) is the reason for the good performance of FastTF. The effects of stacking the data according to its base frequency and the new architecture need to be disentangled (see also (Q2))
- (major) The authors did not provide statistics of their achieved results, e.g. variation of the metrics across multiple runs with different seeds, making it possible to have cherry-picked results (not necessarily the case in reality)
- (minor) The related work could be extended to include additional works such as:
    - Wen, Q., He, K., Sun, L., Zhang, Y., Ke, M., \& Xu, H. (2021, June). RobustPeriod: Robust time-frequency mining for multiple periodicity detection. In Proceedings of the 2021 international conference on management of data (pp. 2328-2337).
    - Wen, Q., Zhou, T., Zhang, C., Chen, W., Ma, Z., Yan, J., \& Sun, L. (2023, August). Transformers in time series: a survey. In Proceedings of the Thirty-Second International Joint Conference on Artificial Intelligence (pp. 6778-6786).
    - A. Weyrauch et al., "ReCycle: Fast and Efficient Long Time Series Forecasting with Residual Cyclic Transformers," 2024 IEEE Conference on Artificial Intelligence (CAI), Singapore, Singapore, 2024, pp. 1187-1194, doi: 10.1109/CAI59869.2024.00212.
- (minor) quote at the beginning of the text seems to be out of place for a brief proceedings article
- (minor) division into patches and downsampling as used in the submission are identical, use only one to avoid confusion (preferably patching since downsampling implies loss of information)
- (minor) the description of the Exchange dataset in Appendix A.5 seems to be mixed up with that of a different dataset
- (minor) the number of parameters given in section 5.3 is correct for the number of complex degrees of freedom; for comparison with models that do not work with complex numbers this is slightly misleading; give additionally the number of real parameters (even if that is just a factor of two)
- (minor) the magnitude of the error implies that all the metrics given are still normalized; either give denormalized metrics or acknowledge that it is still normalized
- (minor) Transformer part of the Related Work section: "Informer and Autoformer capture the temporal dependence of time-series" is non-informative. We would encourage the authors to additionally state how these are captured. Furthermore, "..., while FEDformer models the frequency domain of the time-series.". this implies that Autoformer and Informer do not work in Fourier space, which does not hold for the Autoformer, albeit the motivation is different compared to FEDformer
- (minor) The authors should closely check the manuscript for grammar and phrasing. Some of the minor issues that the reviewer found are:
    - "Natual Correlation" should likely be "Natural correlation" (l. 230)
    - Inconsistencies with the use of capitalization/title case, e.g. "**N**atural **C**orrelation'' vs. "**T**he **e**ffect of **d**ownsampling"
- (minor) The reviewer would like to request from the authors to increase adherence with the paper template, which includes, but is not limited to:
    - Tables need to be centered
    - Figure colors should be legible even on black/white printouts, currently some pastel colors are difficult to read even in the PDF
    - Please use large enough font sizes for all visual elements
    - Refrain from using color to highlight elements in tables, especially the lime green.
- (minor) The reviewer does not agree with the use of "Theorem" as used in this manuscript
       - Theorem 1 is not a new insight by the authors and therefore does not need to be proven again. Theorem would imply that it is novel.
        - Theorem 2 is more of an observation or counting not a mathematical insight

**Questions:**

- What is the energy consumption during training? If convergence of FastTF is faster and the number of parameters is smaller, does this convert into energy savings?
- How does FastTF perform if the patch size $P$ is not a multple or integer divisor of the fundamental frequency, e.g. 24h for ETTh? This could be a stress test for the ability of the SFM mechanism to deal with spectral leakage as claimed.
- The reviewer has observed that FastTF is mostly just learning a singular template pattern that is, if at all, simply shifted by the local mean. While this performs well (see your metrics), it is quite questionable to refer to true learning by the model. What would happen if the model was faced with strong out-of-distribution data, e.g. with strong noise or shifts?

---

### Official Review · Reviewer_R8hR · 2024-11-01

**Soundness:** 3
**Presentation:** 3
**Contribution:** 2
**Rating:** 5
**Confidence:** 3

**Summary:**

This manuscript proposes a lightweight long-term time series prediction model based on time-frequency domain information, which uses the compressibility of frequency domain information to significantly reduce model parameters so that it can be deployed on a wider range of platforms. While maintaining an extremely low number of parameters, the model can still achieve competitive prediction accuracy. The authors conduct a large number of experiments to prove the effectiveness of the method and deploy it on the FPGA platform to demonstrate its extremely low hardware requirements.

**Strengths:**

1. This study reduces the number of model parameters and computational complexity to an extremely low level while maintaining its predictive effectiveness. This is a novel and impressive study.
2. The experiments conducted by the authors are very detailed and reliable, and they provide a detailed analysis of various performance aspects including algorithm complexity and resource usage.
3. The manuscript is well-written and the relevant figures and tables are clear and easy to read.

**Weaknesses:**

1. The practical value and motivation of this research are questionable. The performance of various types of computing hardware is constantly increasing, and whether it is really necessary to reduce the number of parameters to 4 is a question that needs to be considered.
2. The number of experimental datasets used by the study is too small. The main experimental results presented in the manuscript are from only four datasets, which weakens the persuasiveness of the experiment.
3. In the experimental results, the method does not seem to perform as well as other SOTA models on the electricity and traffic datasets.

**Questions:**

1. The performance of the PatchTST provided by the author seems to be quite different from that of the original paper. Although the authors state that it is caused by a code bug, can a single error in drop_last lead to such a large performance gap?
2. Fits（arXiv:2307.03756, 2023.）also proposed a lightweight frequency domain prediction algorithm. What are the similarities and differences between FastTF and it? Can the authors give a detailed comparison to highlight the research contribution?

---

> ### Author Response · Authors · 2024-11-13
> **Author Rebuttal**
>
> **Thank you for your valuable comments and questions. We have carefully considered your feedback and made every effort to address all of your concerns. Below, we provide detailed responses to each of your points, and we hope that our clarifications and revisions meet your expectations.**
>
> >**W1**: The practical value and motivation of this research are questionable. The performance of various types of computing hardware is constantly increasing, and whether it is really necessary to reduce the number of parameters to 4 is a question that needs to be considered.
>
> >**Response:** Thank you for your comments. Indeed, the computational power of modern high-performance hardware continues to grow, and clusters composed of A100 and H100 GPUs are capable of handling the demands of most large-scale models. Still, the following points should be considered:
>
> >1. In many scenarios (such as smart home devices, smartwatches, industrial monitoring, autonomous driving, etc.), there is still a significant need for edge computing and Internet of Things (IoT) devices, as well as embedded systems. These devices often have very limited computational and storage resources (as mentioned in the manuscript, such as FPGA boards and ESP32 devices). They typically face constraints such as low power consumption, limited storage, and high real-time requirements in specific applications (e.g., in the aerospace sector). These limitations make the deployment of small, fast models a critical research direction.
> >2. The computational cost grows with the increase in data volume (as some transformer models exhibit quadratic growth in computational complexity). In complex scenarios (where the number of channels is large), large models can become difficult to deploy at scale, and the associated deployment costs increase sharply (including the cost of hardware, hardware maintenance, and the energy consumption driven by excessive computational demands). Our model not only supports large-scale deployment but also significantly reduces operational costs.
> >3. In scenarios such as autonomous driving systems and power grid management systems, time series prediction may only serve as one module within a larger system. Our model is capable of preserving prediction accuracy while freeing up as much computational resource as possible for other modules, which is vital for the overall system's real-time performance and stability.
>
> >It is important to emphasize that our model **does not sacrifice accuracy for being lightweight**. On the contrary, we have achieved optimal performance in both accuracy and model size.

---

> ### Author Response · Authors · 2024-11-13
>
> >**W2**: The number of experimental datasets used by the study is too small. The main experimental results presented in the manuscript are from only four datasets, which weakens the persuasiveness of the experiment.
>
> >**Response:** Thank you for your valuable feedback. We would like to kindly point out that, as mentioned in the manuscript, we have actually conducted experiments on **five datasets** (instead of four, I think it may be because that the results of the fifth dataset -- the "weather" dataset -- were presented separately.)
> , with detailed results presented in the original main text. Additionally, we have included results from three more datasets in the **Appendix C.4**. We hope this clarifies the concern regarding the number of datasets used, and we believe this provides a comprehensive evaluation of our model's performance. The five datasets in the original manuscript are as follows: **ETTh1, ETTh2, Electricity, traffic, and weather**. The three additional datasets in the Appendix are **ETTm1, ETTm2, and Exchange**.
>
> >The original results of the weather dataset are presented as follows:
>
> | Horizon        | 96 | 192 | 336 | 720 |
> |-------------------|--------------|---------------|---------------|---------------|
> | FEDformer (2022b) | 0.217        | 0.276         | 0.339         | 0.403         |
> | TimesNet (2023)   | 0.172        | 0.219         | 0.280         | 0.365         |
> | PatchTST (2023)   | 0.149      | 0.194         | 0.245         | 0.314         |
> | DLinear (2023)    | 0.176        | 0.218         | 0.262         | 0.323         |
> | FITS (2024)       | 0.145    | 0.188     | 0.236     | 0.308     |
> | TimeMixer (2024)  | 0.147        |0.189     | 0.241       | 0.310       |
> | ModernTCN (2024)  | 0.149        | 0.196         | 0.238         | 0.314         |
> | Koopa (2024)      | 0.154        | 0.193         | 0.245         | 0.321         |
> | MICN (2023)       | 0.161        | 0.220         | 0.278         | 0.311         |
> | **FastTF (ours)** | **0.140**  | **0.180**   | **0.232**   | **0.301**   |
>
> >The additional results of the three datasets in **Appendix C.4** are presented as follows:
>
> | Dataset           | ETTm1 (96) | ETTm1 (192) | ETTm1 (336) | ETTm1 (720) | ETTm2 (96) | ETTm2 (192) | ETTm2 (336) | ETTm2 (720) | Exchange (96) | Exchange (192) | Exchange (336) |
> |-------------------|------------|-------------|-------------|-------------|------------|-------------|-------------|-------------|---------------|----------------|----------------|
> | FEDformer (2022b) | 0.326      | 0.365       | 0.392       | 0.446       | 0.180      | 0.252       | 0.324       | 0.410       | 0.139         | 0.256          | 0.426          |
> | TimesNet (2023)   | 0.338      | 0.371       | 0.410       | 0.478       | 0.187      | 0.249       | 0.321       | 0.497       | 0.107         | 0.226          | 0.367          |
> | PatchTST (2023)   | **0.290**      | **0.332**       | **0.366**       | 0.416       | 0.165      | 0.220       | 0.274       | 0.362       | 0.093         | 0.192          | 0.350          |
> | DLinear (2023)    | 0.299      | 0.335       | 0.369       | 0.425       | 0.167      | 0.224       | 0.281   | 0.397       | 0.081     | **0.157**      | 0.305          |
> | U-Mixer (2024)    | 0.317      | 0.369       | 0.395       | 0.443       | 0.178      | 0.243       | 0.331       | 0.434       | 0.087         | 0.171          | 0.285          |
> | Koopa (2024)      | 0.294  | 0.337       | 0.380       | 0.426       | 0.171  | 0.226       | 0.283       | 0.394       | 0.083         | 0.184          | 0.331          |
> | MICN (2023)       | 0.314      | 0.359       | 0.398       | 0.459       | 0.178      | 0.245       | 0.295       | **0.389**   | 0.102         | 0.172          | **0.272**          |
> | **FastTF (ours)** | 0.302 (4th)| 0.334 (2nd)   | 0.372 (3rd)   | **0.415**   | **0.162**  | **0.215**   | **0.266**       | **0.349**       | **0.080**     | 0.167 (2nd)      | 0.304 (2nd)      |

---

> ### Author Response · Authors · 2024-11-13
> **Author Rebuttal**
>
> >**W3**: In the experimental results, the method does not seem to perform as well as other SOTA models on the electricity and traffic datasets.
>
> >**Response:** Thank you for your feedback. In the original Table 2, our model ranks first in 16 out of the 20 experiments and second in 4 experiments, only behind PatchTST. While our model underperforms the transformer-based PatchTST in a few cases, it's important to emphasize that:
>
> >1. PatchTST requires nearly 20M parameters on these two datasets, while our model achieves the best performance with only 1K and 32K parameters, respectively (detailed results can be found in Appendix Table 10). This represents a reduction of 2–3 orders of magnitude in the number of parameters compared to PatchTST. From this perspective, the authors believe that FastTF is already sufficiently powerful.
> >2. Models based on time-frequency domains are particularly effective at predicting longer horizons. This is because the global perspective of the frequency domain allows for better capture of cross-patch periodic information in stationary sequences. This explains why our model outperforms PatchTST at the 720-point horizon on the Traffic dataset. Furthermore, in other datasets, such as Electricity, we also observe that our model shows the greatest advantage at the 720-point horizon. This ability to make more accurate long-horizon predictions is clearly crucial for long sequence forecasting tasks.
>
> >**Q1**: The performance of the PatchTST provided by the author seems to be quite different from that of the original paper. Although the authors state that it is caused by a code bug, can a single error in drop_last lead to such a large performance gap?
>
> >**Response:** In the previous code, shuffle in the data_loader was set to False and drop_last was set to True, which caused the same last batch in the test data to be dropped (as long as the batch_size remained the same). This batch of test data resulted in slightly better performance for most models on ETTh1 and ETTh2. The results reported in our paper are consistent with [1] and [2], and the reproduction results in these two papers serve as strong evidence for this.
>
> [1] Lin S, Lin W, Wu W, et al. SparseTSF: Modeling Long-term Time Series Forecasting with 1k Parameters[J]. arXiv preprint arXiv:2405.00946, 2024.
>
> [2] Xu Z, Zeng A, Xu Q. FITS: Modeling time series with $10 k $ parameters[J]. arXiv preprint arXiv:2307.03756, 2023.

---

> ### Author Response · Authors · 2024-11-13
> **Author Rebuttal**
>
> >**Q2**: Fits（arXiv:2307.03756, 2023.）also proposed a lightweight frequency domain prediction algorithm. What are the similarities and differences between FastTF and it? Can the authors give a detailed comparison to highlight the research contribution?
>
> >**Response:** In terms of methodology, FITS is essentially a frequency-domain interpolation model. It interpolates the frequency domain from the original $L$-point sequence of the look-back window to an $L+H$-point sequence in the frequency domain, and then obtains the prediction through an inverse transform. In other words, FITS reconstructs both the look-back window and the prediction sequence, but only uses the latter in the loss function. Specifically, the similarities and differences between FITS and FastTF are as follows:
>
> >1. FITS is a pure frequency-domain model, treating the input sequence and the input+output sequence as a whole for frequency-domain interpolation. While this method leverages the global characteristics of the frequency domain, it overlooks the fact that the FFT operation tends to blur time information associated with specific frequency points. To better handle potential changes in frequency values over different time periods, we propose FastTF, a model that fuses both time and frequency domains. Unlike FITS, FastTF does not reconstruct the input+output sequence as a whole. Instead, it predicts the frequency points across patches, using the frequency values from different patches in the look-back window to predict the frequency values in the output sequence across different patches.
> >2. In FITS, the interpolation layer derives the frequency points of the input+output sequence from the input frequency points, whereas in FastTF, the SFM (Spectral Frequency Mixer) mixes the frequency points within each patch without involving interpolation or prediction between different frequency points.
> >3. FITS does not involve any downsampling operations.
> >4. Both FITS and FastTF use $f_c$ for frequency-domain filtering, which is a common operation in signal processing. However, in FastTF, since the downsampling operation decreases the upper limit of frequency representation, the purpose of filtering is less prominent. The main reason for filtering in FastTF is to ensure that the number of frequency points can be divided evenly by the sparse grouping number $K$.
>
> >In summary, although both FITS and FastTF utilize the rFFT operation, their fundamental principles and motivations are quite different. Additionally, in FastTF, we employ techniques such as cross-patch weight sharing, weight sparsification, and cross-frequency-point weight sharing to reduce the number of parameters, which the authors believe is another key innovation of FastTF.
>
> **Thank you again for your valuable time. We hope our response has addressed your concerns. If you have any further questions, please feel free to let us know!**

---

### Official Review · Reviewer_Mfsg · 2024-11-03

**Soundness:** 2
**Presentation:** 3
**Contribution:** 3
**Rating:** 1
**Confidence:** 4

**Summary:**

This paper introduces FastTF, a lightweight model for long-term time series forecasting that operates in the time-frequency domain. The key innovation is achieving strong predictive performance with remarkably few parameters - as few as 4 parameters in certain configurations.
The paper develops a novel architecture that combines patch-wise downsampling for weight sharing, a Sparse Frequency Mixer to capture correlations between frequency points, and a patch predictor to forecast temporal variations. The authors provide theoretical foundations for their design choices, drawing on the Nyquist sampling theorem and analysis of spectral properties.
Through extensive experiments across multiple datasets, FastTF demonstrates competitive or superior performance compared to state-of-the-art models while using orders of magnitude fewer parameters. The authors also show successful deployment on FPGA hardware with low resource usage and latency, making it particularly suitable for resource-constrained applications.
The work represents a significant step toward efficient time series forecasting, offering a solution that is both lightweight enough for edge devices and accurate enough for practical applications.

**Strengths:**

Originality:
The paper shows originality in that rather than pursuing better accuracy through larger models, it takes the novel approach of extreme model compression while maintaining performance.
Quality:
The technical quality is high, with theoretical foundations and empirical validation. The authors provide thorough mathematical analysis, including proofs related to sampling theory and spectral properties. The experimental evaluation is ok, covering multiple datasets, and horizons. The ablation studies and hyperparameter analyses demonstrate robustness. Notably, the authors went beyond software simulation to validate their approach on actual FPGA hardware, providing practical evidence of deployability. The comparison with numerous baselines across different model families (Transformers, CNNs, MLPs) strengthens the findings.
Clarity:
The paper is well-structured and clearly written. Complex technical concepts are explained with appropriate mathematical rigor while maintaining readability. The authors use effective visualizations to illustrate key concepts like spectral leakage and frequency correlations.
Significance:
The work's significance is good in both theoretical and practical terms. Theoretically, it demonstrates that extremely lightweight models can match or exceed the performance of much larger models in time series forecasting, challenging conventional wisdom about model capacity requirements. Practically, the ability to deploy effective forecasting models on resource-constrained devices opens up new applications in edge computing and IoT scenarios. The dramatic reduction in parameter count (up to 46,400x fewer than some baselines) while maintaining competitive performance represents a significant advance in efficient deep learning.

**Weaknesses:**

Dataset Dependency: The headline achievement of 4-parameter model works well on ETTh1 but requires orders of magnitude more parameters on other datasets (1928 for ETTh2, 4329 for Electricity). This variation isn't well explained and suggests important dataset dependencies not fully explored.

Missing Analysis: The paper doesn't adequately explore when the model might fail or what dataset characteristics lead to optimal performance. Including comparison with recent lightweight approaches like SparseTSF would better contextualize the contribution.

TimeMixer shows in Table 3 but not Table 2, causing some concerns.

**Questions:**

TimeMixer is missing from Table 2. Was it intentionally omitted from Table 2 or if this was an oversight? Including TimeMixer in Table 2 would provide a more comprehensive comparison across all datasets and ensure consistency with Table 3

---

> ### Comment · Reviewer_Mfsg · 2024-12-03
>
> I want to thank the authors for all their efforts in this paper. I have finalized my scores after considering my interactions with the authors during the discussion period. Since some major concerns were not addressed, I updated my score accordingly.

---

### Official Review · Reviewer_Ur6Z · 2024-11-04

**Soundness:** 2
**Presentation:** 2
**Contribution:** 2
**Rating:** 3
**Confidence:** 5

**Summary:**

This paper introduces a lightweight model, FastTF, which utilizes only 4 parameters. Leveraging the global characteristics and compressibility of information in the frequency domain of time series, this model captures key information through patch-wise downsampling, Sparse Frequency Mixer (SFM), and patch predictor. The experimental results are used to try to demonstrate the effectiveness of FastTF. The experimental results are used to try to demonstrate the effectiveness of FastTF.

**Strengths:**

1. The author's focus on long-term time series forecasting issues is worthy of research.

2. The model exhibits good motivation and innovation.

**Weaknesses:**

1. The article lacks research on important references, such as the author's focus on FITS and ModernTCN at ICLR 2024, but does not consider contemporaneous models like iTransformer (Attention-based)[1]. At the same time, the author overlooks earlier methods, such as Basisformer (attention-based)[2] and WITRAN (RNN-Based)[3] presented at NeurIPS 2023 and FiLM[4] at NeurIPS 2022.

[1] Liu, Y., Hu, T., Zhang, H., Wu, H., Wang, S., Ma, L., & Long, M. (2024). iTransformer: Inverted Transformers Are Effective for Time Series Forecasting. In The Twelfth International Conference on Learning Representations.

[2] Ni, Z., Yu, H., Liu, S., Li, J., & Lin, W. (2023). BasisFormer: Attention-based Time Series Forecasting with Learnable and Interpretable Basis. In Thirty-seventh Conference on Neural Information Processing Systems.

[3] Jia, Y., Lin, Y., Hao, X., Lin, Y., Guo, S., & Wan, H. (2023). WITRAN: Water-wave Information Transmission and Recurrent Acceleration Network for Long-range Time Series Forecasting. In Thirty-seventh Conference on Neural Information Processing Systems.

[4] Zhou, T., Ma, Z., Wang, X., Wen, Q., Sun, L., Yao, T., ... & Jin, R. (2022). FiLM: frequency improved legendre memory model for long-term time series forecasting. In Thirty-sixth Conference on Neural Information Processing Systems.

2. The experiments are insufficient. The baselines mentioned in W1 were not compared in this paper. Furthermore, I noticed that in Table 2, the authors did not compare TimeMixer and ModernTCN across the four datasets, and in Table 3, U-Mixer was not included in the comparison. Therefore, the conclusion that FastTF achieves SOTA is not fully supported.

3. The absence of code publication results in a lack of reproducibility.

4. The reported results in the paper are questionable. Experiment results can vary across different platforms, and to ensure fairness, when platforms are not consistent, all models should undergo parameter search (determining the best parameters using a validation set). Otherwise, it is difficult to guarantee the validity of the experimental outcomes on the current platform. Unfortunately, I found that the authors might not have conducted this work. For example, the author used an NVIDIA RTX 4090 GPU while MICN used an NVIDIA RTX A5000 24GB GPU. However, the reported results by the author are highly similar to those of MICN, which is highly questionable. Based on this, it is also challenging to support the conclusions drawn in the paper.

**Questions:**

1. Can the author include the methods mentioned in Weakness 1 and 2 in the experiments on all datasets?

2. Can the author, based on Question 1, describe more detailed baseline model parameter search results (such as e_layers, d_models, n_heads, etc.)? While model lightweighting is commendable, performance is more crucial than efficiency. I believe that by conducting more thorough experiments to demonstrate this, the quality of the paper can be enhanced.

---

### Official Review · Reviewer_r1r1 · 2024-11-04

**Soundness:** 3
**Presentation:** 2
**Contribution:** 2
**Rating:** 3
**Confidence:** 4

**Summary:**

This paper leverages the global nature and information compression capabilities of time series data in the frequency domain, proposing a powerful yet lightweight model for long-term time series forecasting. Specifically, FastTF includes three key components: patch-wise downsampling, Sparse Frequency Mixer (SFM), and a patch predictor to capture temporal variations in frequency components across different patches.

**Strengths:**

1. The authors propose a lightweight model that can be deployed on resource-constrained devices.
2. FastTF combines the global perspective and the information compression capabilities of the frequency domain.
3. FastTF is demonstrated to be effective, achieving state-of-the-art (SOTA) performance in the experiments.

**Weaknesses:**

1. The related work section is insufficient. More lightweight methods should be included, such as TSLANet, SparseTSF, TSMixer, and even lightweight Large Time Series TTM, to highlight the challenges this paper seeks to address.

[1] Eldele E, Ragab M, Chen Z, et al. TSLANet: Rethinking Transformers for Time Series Representation Learning[C]//Forty-first International Conference on Machine Learning, 2024.

[2] Lin S, Lin W, Wu W, et al. SparseTSF: Modeling Long-term Time Series Forecasting with* 1k* Parameters[C]//Forty-first International Conference on Machine Learning, 2024.

[3] Ekambaram V, Jati A, Nguyen N, et al. Tsmixer: Lightweight mlp-mixer model for multivariate time series forecasting[C]//Proceedings of the 29th ACM SIGKDD Conference on Knowledge Discovery and Data Mining. 2023: 459-469.

[4] Vijay E, Jati A, Dayama P, et al. Tiny Time Mixers (TTMs): Fast Pre-trained Models for Enhanced Zero/Few-Shot Forecasting of Multivariate Time Series[J]. arXiv, 2024.

2. The explanation of "spectral leakage" is unclear. While it is an existing problem, the explanation regarding spectral leakage between frequency points is difficult to understand. Please clarify this further to improve reader comprehension of the motivation.

3. Regarding the search for optimal hyperparameters in FastTF: The search for optimal hyperparameters in FastTF involves four hyperparameters: PS, M, f_c, and K. Is there any systematic approach to find the optimal values, or is it purely empirical? If it's empirical, it could be time-consuming and require significant computational resources.

4. It is suggested that the authors include additional lightweight methods for comparison in the experiment section. Examples include TSLANet, SparseTSF, TSMixer, and TTM, which would help highlight the innovation and effectiveness of FastTF.

[1] Eldele E, Ragab M, Chen Z, et al. TSLANet: Rethinking Transformers for Time Series Representation Learning[C]//Forty-first International Conference on Machine Learning, 2024.

[2] Lin S, Lin W, Wu W, et al. SparseTSF: Modeling Long-term Time Series Forecasting with* 1k* Parameters[C]//Forty-first International Conference on Machine Learning, 2024.

[3] Ekambaram V, Jati A, Nguyen N, et al. Tsmixer: Lightweight mlp-mixer model for multivariate time series forecasting[C]//Proceedings of the 29th ACM SIGKDD Conference on Knowledge Discovery and Data Mining. 2023: 459-469.

[4] Vijay E, Jati A, Dayama P, et al. Tiny Time Mixers (TTMs): Fast Pre-trained Models for Enhanced Zero/Few-Shot Forecasting of Multivariate Time Series[J]. arXiv, 2024.

5. Can the authors compare it with time-frequency methods, such as JTFT and TFDNet? I am concerned about whether relying solely on frequency will affect performance.

[1] Chen Y, Liu S, Yang J, et al. A Joint Time-Frequency Domain Transformer for multivariate time series forecasting[J]. Neural Networks, 2024, 176: 106334.

[2] Luo Y, Lyu Z, Huang X. TFDNet: Time-Frequency Enhanced Decomposed Network for Long-term Time Series Forecasting[J]. arXiv preprint arXiv:2308.13386, 2023.

6. The most critical issue is that two claims made in the introduction—lightweight and interpretability—have not been experimentally verified. In particular, which aspects of the design contribute to interpretability? How can this be verified in the experimental section?

**Questions:**

See Weaknesses

---

### Meta-Review · Area_Chair_g8LA · 2024-12-20

**Metareview:**

This paper introduces a lightweight model called FastTF, leveraging the time-frequency domain for long-term time series forecasting. Specifically, FastTF includes three key components, namely, patch-wise downsampling, Sparse Frequency Mixer (SFM), and a patch predictor to capture temporal variations in frequency components across different patches. However, significant weaknesses and concerns were raised by reviewers as follows.

**Insufficient Literature Coverage**: Recent methods for time series forecasting and lightweight methods are not well covered in the related work section.

**Experimental Limitations**: The evaluation lacks critical baseline comparisons and uses only a limited number of datasets.

**Unverified Claim about Interpretability**: The claimed interpretability of the model is not supported with sufficient evidence or analysis.

During the rebuttal, the authors responded to the comments from one reviewer only, while did not address the comments from the other reviewers. Considering those significant concerns not addressed by the authors, I would like to recommend rejecting this paper.

**Additional Comments On Reviewer Discussion:**

The authors responded to the comments from Reviewer R8hR only; however, the concern regarding the limited number of experimental datasets was not adequately addressed. Additionally, the authors opted not to respond to the comments raised by the other reviewers, leaving several critical issues unresolved.

---

### Decision · Program_Chairs · 2025-01-22

Reject